



# Quantification of CH4 emissions from waste disposal sites near the city of Madrid using ground- and space-based observations of COCCON, TROPOMI and IASI

Qiansi Tu[1], Frank Hase[1], Matthias Schneider[1], Omaira García[2], Thomas Blumenstock[1], Tobias Borsdorff[3], Matthias Frey[1,a], Farahnaz Khosrawi[1], Alba Lorente[3], Carlos Alberti[1], Juan J. Bustos[2], Andre Butz[4], Virgilio Carreño[2], Emilio Cuevas[2], Roger Curcoll[5,b], Christopher J. Diekmann[1], Darko Dubravica[1], Benjamin Ertl[1,6], Carme Estruch[5], Sergio Fabián León-Luis[2], Carlos Marrero[2], Josep-Anton Morgui[5], Ramón Ramos[2], Christian Scharun[1], Carsten Schneider[4], Eliezer Sepúlveda[2], Carlos Toledano[7], Carlos Torres[2]

[1] Karlsruhe Institute of Technology (KIT), Institute of Meteorology and Climate Research (IMK-ASF), Karlsruhe, Germany
[a] now at National Institute for Environmental Studies, Tsukuba, Japan
[2] Izaña Atmospheric Research Centre (IARC), Meteorological State Agency of Spain (AEMet), Tenerife, Spain
[3] SRON Netherlands Institute for Space Research, Utrecht, the Netherlands
[4] Institut für Umweltphysik, Heidelberg University (UH), Germany
[5] Institut de Ciència i Tecnologia Ambientals (ICTA) - Autonomous University of Barcelona, Spain
[b] now at Institute of Energy Technologies, Polytechnic University of Catalonia, Barcelona, Spain
[6] Karlsruhe Institute of Technology, Steinbuch Centre for Computing (SCC), Karlsruhe, Germany
[7] Group of Atmospheric Optics, University of Valladolid, Spain

*Correspondence to*: Qiansi Tu (qiansi.tu@kit.edu)

**Abstract.** The objective is to derive methane ($CH_4$) emissions of the metropolitan city Madrid Spain from the $CH_4$ enhancements seen by the space-borne and the ground-based instruments. This study applies satellite-based measurements from the TROPOspheric Monitoring Instrument (TROPOMI) and the Infrared Atmospheric Sounding Interferometer (IASI) together with measurements from the ground-based COllaborative Carbon Column Observing Network (COCCON) instruments.

In 2018, a two-week field campaign for measuring the atmospheric concentrations of greenhouse gases was performed in Madrid in the framework of Monitoring greenhousE Gas EmIssions of Madrid city (MEGEI-MAD) project. Five COCCON instruments were deployed at different locations around the Madrid city center enabling the observation of total column averaged $CH_4$ mixing ratios ($XCH_4$). Using available wind data, the differences between $CH_4$ columns observed at these locations allow to estimate the emissions emerging from the surrounded area. In addition, based on the dominating wind direction in the Madrid region, we calculate the difference of the satellite data maps for two opposite wind regimes (northeast – southwest, NE – SW). In the following, we refer to the resultant signal as the wind-assigned anomaly. We use TROPOMI tropospheric nitrogen dioxide ($NO_2$) observations as a test to verify our method of wind-assigned anomaly and its implementation, taking advantage of the much better detectability of the plume due to the short lifetime and low background concentrations of $NO_2$. Pronounced bipolar plumes are found along NE and SW wind direction, which implies that our method





of wind-assigned anomaly is working as expected. The wind-assigned TROPOMI $XCH_4$ anomaly shows much weaker symmetric plumes than $NO_2$ due to the long lifetime of $CH_4$ and in consequence a high accumulated background of $CH_4$ in the atmosphere. The wind-assigned plume method is also applied to the tropospheric and upper tropospheric/stratospheric column averaged $CH_4$ mixing ratio products (in the following referred to as $TXCH_4$ and $UTSXCH_4$) derived from a-posteriori merged Infrared Atmospheric Sounding Interferometer (IASI) profile and TROPOMI total column data.

Based on the NE and SW wind fields, we developed a simple plume model locating the source at three waste disposal sites east of Madrid for $CH_4$. As $CH_4$ emission strength we estimate $7.4\times10^{25} \pm 6.4\times10^{24}$ molec $s^{-1}$ from the TROPOMI $XCH_4$ data and $7.1\times10^{25} \pm 1.0\times10^{25}$ molec $s^{-1}$ from the TROPOMI&IASI merged $TXCH_4$ data. The COCCON observations indicate a weaker $CH_4$ emission strength of around $3.7\times10^{25}$ molec $s^{-1}$ from local source (near to the Valdemingómez waste plant) in accordance with observations in a single day and. All emission rates estimated from the different observations are significantly

larger than the emission rates provided via the official Spanish Register of Emissions and Pollutant Sources.

## 1 Introduction

Methane ($CH_4$) is the second most important anthropogenic greenhouse gas (GHG) after carbon dioxide ($CO_2$) and contributes about 23.4% to the radiative forcing by long-lived GHGs in the atmosphere (Etminan et al., 2016). The amount of atmospheric $CH_4$ increased to 1880 ppb in 2019, corresponding to 260% of the pre-industrial level (World Meteorological Organization,

2020). The global atmospheric $CH_4$ sources are approximately 40% of natural sources (e.g. wetlands and termites) and about 60% of anthropogenic sources (e.g. agriculture and fossil fuels, Saunois et al., 2020). $CH_4$ is primarily removed through the reaction with the hydroxyl radicals (OH), mostly in the troposphere, which accounts for about 90% of the global $CH_4$ sink (Kirschke et al., 2013). Consequently, small changes in OH can lead to considerable variability in $CH_4$ amounts (Dlugokencky et al., 2011). It is therefore important to increase our knowledge on how the different sources and sinks affect the $CH_4$ amount

in the atmosphere. Understanding the sources and sinks of $CH_4$ is also of importance for future climate emission scenarios. However, $CH_4$ sources and sinks are still not fully understood. Although a lot of research studies had their focus on the global OH sink of $CH_4$ and the sum of $CH_4$ sources, which are relatively well known, large uncertainties still remain in each of the individual $CH_4$ sources (De Wachter et al., 2017). The $CH_4$ emitted from wetlands and other inland waters are the most important uncertainty components of the $CH_4$ budget (Saunois et al., 2016).

Satellite observations of $CH_4$ started with the launch of the Interferometric Monitor for Greenhouse gases (IMG) aboard the ADEOS satellite in August 1996 (Clerbaux et al., 2003). From 2002-2012 the SCIAMACHY (Scanning Imaging Absorption Spectrometer for Atmospheric Cartography) on board the European Envisat satellite performed measurements of total column $CH_4$ (Frankenberg et al., 2006). The Greenhouse Gases Observing Satellite (GOSAT) launched in 2009 is the first satellite dedicated to the monitoring of atmospheric GHGs and is still in operation (Kurze et al., 2009). Atmospheric $CH_4$ measurements

from satellite instruments have been used to study $CH_4$ hotspot emission (e.g. the anomalous $CH_4$ emission source regions (Kort et al., 2014), anthropogenic emissions (Marais et al., 2014), and tropical wetlands (Lunt et al., 2019)). A lot of research





has also been carried out to map emission trends (Schneising et al., 2014; Maasakkers et al., 2019) and to estimate regional emissions (Monteil et al., 2013; Turner et al., 2015; Kuze et al., 2020; Tunnicliffe et al., 2020). The current global GOSAT observations are of high quality but have sparse spatial and temporal coverage, limiting the capability to estimate the changes

in daily emissions on small scales (Lorente et al., 2021).

Launched in October 2017, the TROPOspheric Measuring Instrument (TROPOMI) on board the Copernicus Sentinel-5 Precursor satellite provides complete daily global coverage of $CH_4$ with an unprecedented resolution of $7 \times 7$ km². The resolution was upgraded to $5.5 \times 7$ km² in August 2019. The TROPOMI instrument is therefore able to map the $CH_4$ enhancements due to emissions on fine scale and to detect large point sources (Varon et al., 2019). Borsdorff et al. (2020) also

investigated the CO emissions of the metropolis Mexico City using TROPOMI observations and the study showed that TROPOMI has the potential to constrain the emission strengths on regional area. TROPOMI $CH_4$ data show an excellent agreement with the measurements from the validated GOSAT (Hu et al., 2018) and the ground-based Total Carbon Column Observing Network (TCCON) (Lorente et al., 2021).

Satellite retrievals using thermal infrared nadir spectra as observed for instance by IASI or TES (Tropospheric Emission

Spectrometer) are especially sensitive to $CH_4$ concentrations between the middle troposphere and the stratosphere (e.g. Siddans et al., 2017; Garcia et al., 2017; De Wachter et al, 2017; Kulawik et al., 2021; Schneider et al., 2021a). The IASI sensors are currently orbiting aboard of three different Metop (Meteorological operational) satellites and offer twice daily global coverage with high horizontal resolution (ground pixel diameter at nadir is 12 km). The IASI $CH_4$ products have a particular good quality and sensitivity between the middle troposphere and the stratosphere as documented in different validation studies (e.g. Siddans

et al., 2017; De Wachter et al., 2017; García et al., 2018; Schneider et al., 2021a).

TCCON measures solar absorption spectra in the near infrared region by using high-resolution Fourier Transform Infrared (FTIR) spectrometers (Bruker 125HR, Washenfelder et al., 2006), and is primarily designed to provide accurate and long-lasting time series of column-average dry-air molar fractions of GHGs and other atmospheric constituents (Wunch et al., 2011). Therefore, TCCON 125HR provides crucial validation resources for satellite greenhouse gas data, showing TROPOMI $CH_4$

to be of good quality (Hasekamp et al., 2019; Lorente et al., 2021). However, there are to our knowledge no studies on using TROPOMI together with ground-based portable FTIR spectrometer to derive $CH_4$ emission from metropolitan city centers (Pandey et al., 2019; Varon et al., 2019; Gouw et al., 2020). Recently, TCCON 125HR GHG observations have been extended by the COllaborative Carbon Column Observing Network (COCCON, Frey et al., 2019), which is a research infrastructure using well-calibrated low-resolution FTIR spectrometers (EM27/SUN, Gisi et al, 2012) and a common data analysis scheme.

Due to the ruggedness of the portable devices used and simple operability, COCCON is well suited for implementing arrays of spectrometers for the quantification of local GHG sources (Hase et al., 2015; Luther et al., 2019; Vogel et al., 2019; Dietrich et al., 2021).

In this study we analyze nearly three years of TROPOMI total column-average dry-air molar fraction of $CH_4$ ($XCH_4$) measurements together with COCCON spectrometer observations in the framework of the Monitoring greenhousE Gas

EmIssions of Madrid (MEGEI-MAD) project (García et al., 2019), in an attempt of quantifying the $CH_4$ emissions of the most



important metropolitan area of Spain. Section 2 describes our methodology. The results of our study are presented and discussed in Section 3. The conclusions drawn from these results of this study are given in Section 4.

## 2 Method

### 2.1 Ground-based and space-borne instrumentations

#### 2.1.1 COCCON XCH$_4$ data set

The Bruker EM27/SUN is a robust and portable FTIR spectrometer, operating at a medium spectral resolution of 0.5 cm$^{-1}$. The EM27/SUN FTIR spectrometer has been developed by the Karlsruhe Institute of Technology (KIT) in cooperation with Bruker Optics GmbH for measuring GHG concentrations (Gisi et al., 2012; Hase et al., 2016). An InGaAs (Indium-Gallium-Arsenide) photodetector is used as the primary detector, covering a spectral range of 5500 – 11000 cm$^{-1}$. A decoupling mirror reflects
40% of the incoming converging beam to an extended InGaAs photodetector element, covering the spectral range of 4000 – 5500 cm$^{-1}$ for simultaneous carbon monoxide (CO) observations. The recording time, for a typical measurement consisting of five forward and five backward scans, is about 58 seconds in total.

Several successful field campaigns and long-term deployments have demonstrated that the Bruker EM27/SUN FTIR spectrometer is an excellent instrument with good quality, robustness and reliability and its performance offers the potential
to support TCCON (Frey et al., 2015 and 2019; Klappenbach et al., 2015; Chen et al., 2016; Butz et al., 2017; Sha et al., 2019; Jacobs et al., 2020; Tu et al., 2020a and 2020b; Dietrich et al., 2021). The Bruker EM27/SUN spectrometers have become commercially available from April 2014 onwards and currently about 70 spectrometers are operated by different working groups in Germany, France, Spain, Finland, Romania, USA, Canada, UK, India, Korea, Botswana, Japan, China, Mexico, Brazil, Australia and New Zealand. The development of the COCCON (https://www.imk-asf.kit.edu/english/COCCON.php)
became possible by continued European Space Agency (ESA) support. COCCON intends to become a supporting infrastructure for GHG measurements based on common standards and data analysis procedures for the EM27/SUN (and spectrometers of comparable characteristics) (Frey et al., 2019).

Every COCCON instrument is checked (alignment and instrument line shape) and calibrated with respect to a co-located TCCON spectrometer and the primary Bruker EM27/SUN spectrometer unit operated permanently at KIT before deployment
(Frey et al., 2019). For the purpose of COCCON data analysis procedures, a preprocessing tool (PREPROCESS) is applied to the raw interferograms for the generation of spectra and a non-linear least-squares fitting algorithm (PROFFAST) is used for the determination of the desired trace gas abundances from pre-generated spectra. These data processing and analysis tools are open source and freely available (https://www.imk-asf.kit.edu/english/COCCON.php). The ESA provided support for the code development in the framework of projects (COCCON-PROCEEDS and COCCON-PROCEEDS II) and also supports in this
framework the buildup of a central facility of the network. The demonstration of central facility functionalities is performed





by KIT in cooperation with other European partners (e.g. the Belgian Institute for Space Aeronomy and Norwegian Institute for Air Research).

All the Bruker EM27/SUN spectrometers used in the MEGEI-MAD project were operated in accordance with COCCON requirements. The resulting $XCH_4$ data used in this work were generated by the central facility operated by KIT for demonstrating a centralized data retrieval for the COCCON network. For these reasons, we refer to the Bruker EM27/SUN spectrometers as COCCON spectrometers in the following. The COCCON $XCH_4$ data product is derived from the co-observed total column amounts of $CH_4$ and oxygen ($O_2$), and the assumed dry-air molar fraction of $O_2$ (0.2095) (Wunch et al., 2015).

$$XCH_4 = \frac{column_{CH_4}}{column_{O_2}} \times 0.2095$$
**Eq. 1**

### 2.1.2 TROPOMI XCH₄ data set

The TROPOMI data processing deploys the RemoTeC algorithm (Butz et al., 2009, 2011; Hasekamp and Butz, 2008) to retrieve $XCH_4$ from TROPOMI measurements of sunlight backscattered by the Earth's surface and atmosphere in the near-infrared (NIR) and shortwave-infrared (SWIR) spectral bands (Hu et al., 2016, 2018; Hasekamp et al., 2019; Landgraf et al., 2019). This algorithm has been extensively used to derive $CH_4$ and $CO_2$ from GOSAT (Butz et al, 2011; Guerlet et al., 2013). The TROPOMI $XCH_4$ is calculated from the $CH_4$ vertical sub-columns $x_i$ and the dry-air column obtained from the surface pressure from European Centre for Medium-Range Weather Forecasts (ECMWF), and the altitude from the Shuttle Radar Topography Mission (SRTM) (Farr et al., 2007) digital elevation map with a resolution of 15 arcsec (Lorente et al., 2021) analysis:

$$XCH_4 = \sum_{i=0}^{n} \frac{x_i}{column_{dryair}}$$
**Eq. 2**

This study uses the TROPOMI data set of $XCH_4$ between April 30, 2018 and December 30, 2020 within the rectangular area of 39.5°N – 41.5°N and 4.5°W – 3.0°W (125 km × 220 km) over Madrid. In this study we apply a strict quality control to TROPOMI $XCH_4$ (quality value q = 1.0) to exclude data of questionable quality and to assure data under clear-sky and low-cloud atmospheric conditions (Lorente et al., 2021).

### 2.1.3 IASI CH₄ data and its synergetic combination with TROPOMI data

Here we use the IASI $CH_4$ product as generated by the latest MUSICA IASI processor version (Schneider et al., 2021b). Combing these IASI profile data with the TROPOMI total column data causes strong synergies. Schneider et al. (2021a) developed an a posteriori method for such synergetic combination and documented the possibility to detect tropospheric partial column averaged dry-air molar fractions of $CH_4$ ($TXCH_4$) independently from the upper tropospheric/stratospheric dry-air molar fractions of $CH_4$ ($UTSXCH_4$). This is not possible by either the TROPOMI or IASI product individually. In this study we use a tropospheric product averaged from ground to 7 km a.s.l. and an upper tropospheric/stratospheric product averaged from 7 to 20 km a.s.l..



## 2.2 COCCON Madrid campaign

Madrid has almost 3.3 million inhabitants with a metropolitan area population of approximately 6.5 million. Madrid is located on the southern Meseta Central and 60 km south of the Guadarrama mountains with a considerable altitude difference across the city, ranging from 570 to 700 m a.s.l.

This work was made in the framework of the MEGEI-MAD project (García et al., 2019), which aimed to measure atmospheric concentrations of GHGs in an urban environment combing FTIR instruments and ground-level analyszers.

Another objective of MEGEI-MAD was to analyze the possible use of portable COCCON instruments to shape an operational network for Madrid in the future. The MEGEI-MAD project was initiated by the Izaña Atmospheric Research Center (AEMet), in cooperation with two German research groups – the Karlsruhe Institute of Technology and the University of Heidelberg, and two Spanish research groups – the Autonomous University of Barcelona and the University of Valladolid.

Within MEGEI-MAD, a two-week field campaign was carried out from September 24 to October 7, 2018 in Madrid, where

five COCCON instruments were located at five different places circling the metropolitan area (see Figure 1). Table 1 summaries the coordinates, altitudes of the COCCON locations and auxiliary meteorological data collected for data analysis of the observations. The locations have been chosen by considering the prevailing winds and the emssion sources of $CO_2$ and $CH_4$, as well as other technical and logistic criteria (García et al., 2019; García et al., 2021, in preparation).

Though every COCCON instrument is already calibrated at KIT, long-distance transportation and long-term usage under

different conditions might cause some drifts of instrumental characteristics. To minimize any systematic errors due to drifts of the calibration, a pre-campaign side-by-side comparison of the five COCCON instruments used in MEGEI-MAD was carried out at the AEMet Headquarter between September 17 and September 20, 2018, to obtain updated calibration factors (Table 2). The calibration factors are computed as ratios between the observations from each instrument and the ensemble average. These $XCH_4$ calibration factors are in good agreement with previous calibration results for these instruments obtained during side-

by-side comparisons at KIT (empirical standard deviation of the ratios between previous calibration and campaign results ≈ 0.17%). These small changes among the COCCON instruments demonstrate the excellent characteristic of stability. The set of five COCCON spectrometers used in this work is calibrated based on the inter-comparison measurements and the solar zenith angle (SZA) range is limited to 75° to reduce airmass-dependent effects. The instrument-specific calibration factors systematic differences are within ± 0.06 % for $XCH_4$.





**Table 1: Locations of the five COCCON instruments and meteorological records for the MEGEI-MAD field campaign during September 24 – October 7, 2018.**

| Station | EM27/SUN | Latitude (°N) | Longitude (°W) | Altitude (m a.s.l.) | Meteorological Records |
|---|---|---|---|---|---|
| Tres Olivos | KIT SN53 | 40.499 | 3.689 | 736 | Datalogger from AEMet Barajas Airport |
| Barajas | AEMet SN85 | 40.465 | 3.581 | 637 | Barajas Airport |
| Jose Echegaray | DLR SN69 | 40.379 | 3.613 | 633 | Datalogger from DLR Cuatro Vientos Airport |
| Cuatro Vientos | KIT SN52 | 40.368 | 3.780 | 703 | Cuatro Vientos Airport |
| AEMET | KIT SN81 | 40.452 | 3.724 | 685 | AEMET Headquarter |

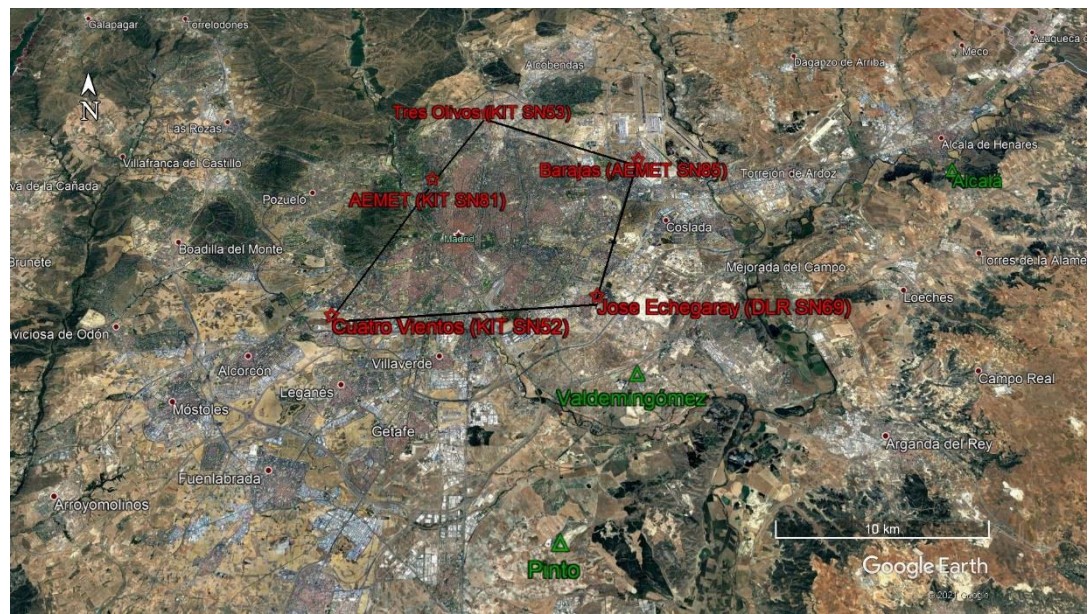


**Figure 1: Locations of the five COCCON instruments used in the Madrid field campaign during September 24 – October 7, 2018, represented with red stars and locations of three waste treatment and disposal plants, represented with the green triangles (© Google Earth).**

**Table 2: XCH₄ calibration factors for the five COCCON instruments from the pre-campaign inter-comparison measurements at the AEMet Headquarter.**

| | KIT SN53 | AEMET SN85 | DLR SN69 | KIT SN52 | KIT SN81 |
|---|---|---|---|---|---|
| side-by-side at AEMet | 0.999212 | 1.000719 | 1.000594 | 1.000077 | 0.999398 |

**2.3 Emission strength calculation using a simple plume model**

The daily plume is modelled as a function of wind direction and wind speed. The schematic dispersion model for describing emissions assumes an expanding cone-shaped plume with the tip at the plume source at location (0,0). The plume cone has an





opening angle of size $\alpha$ and any grid cell within the cone is affected by the emission (see Figure 2). The angle $\alpha$ is a technical

parameter to schematically describe a spreading of the plume and is empirically adjusted to a value of 60°. Different opening

angles are modelled and presented in Figure A- 1. The modelled plume has the most similar shape compared to the TROPOMI

measured $NO_2$ plume (see Section 3.2) when $\alpha >= 60°$. If the grid cell $(x, y)$ locates inside the cone, the column enhancement

for this cell can be calculated by:

$$\Delta colunm_{(x,y)} = \frac{\varepsilon}{v \cdot d_{(x,y)} \cdot \alpha} \qquad \text{Eq. 3}$$

where $\varepsilon$ is the emission strength at the source point in molec s$^{-1}$, $v$ is the wind speed in m s$^{-1}$, $d$ is the distance between the

downwind point and the source, $\alpha$ is the opening angle of the plume in rad (here assumed to be 60°).

The distance from a general grid cell $(x, y)$ from the source is:

$$d(x, y) = \sqrt{x^2 + y^2} \qquad \text{Eq. 4}$$

The enhanced dry-air volume mixing ratio for target species ($\Delta$XVMR) at the center of the grid cell $(x, y)$ can then be

calculated by dividing the column enhancement by the total column of dry air ($column_{dryair}$):

$$\Delta XVMR = \frac{\Delta colunm_{(x,y)}}{column_{dryair}} \qquad \text{Eq. 5}$$

The $column_{dryair}$ is computed from the surface pressure:

$$column_{dryair} = \frac{P_s}{m_{dryair} \cdot g(\varphi)} - \frac{m_{H_2O}}{m_{dryair}} \cdot column_{H_2O} \qquad \text{Eq. 6}$$

where $P_s$ is the surface pressure, $m_{dryair}$ and $m_{H_2O}$ are the molecular masses of dry air ($\sim 28.96\,\text{g} \cdot \text{mol}^{-1}$) and water vapor

($\sim 18\,\text{g} \cdot \text{mol}^{-1}$), respectively. $column_{dryair}$ and $column_{H_2O}$ are the total column amount of dry air and water vapor, and

$g(\varphi)$ is the latitude-dependent surface acceleration due to gravity.

The averaged enhancement of XVMR (plume) over the study area is computed for the selected wind sector. The plume for

the opposite wind regime is also constructed in the same manner. The differences between these two data sets are therefore the

wind-assigned anomalies (see Sect. 3.3). By fitting the modelled wind-assigned anomalies to the anomalies as observed by the

satellite, we can estimate the actual emission strength (see Sect. 2.5). Note that the applied calculation scheme would also be

extendible to areal sources by superimposing such calculations using different locations of the origin.

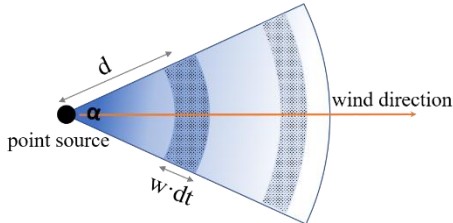

**Figure 2: Sketch of the simple plume model used to explain the CH₄ emission estimation method. The methane at the point source is distributed along the wind direction (wind speed: $v$) in the cone-shaped area with an opening angle of $\alpha$. The point source emits the methane at an emission rate of $\varepsilon$. We assumed the methane molecules are evenly distributed in the dotted area A, and the distance from area A to the point source is d. Therefore, the emitted methane in dt time period equals to the amount of methane in the area A. It yields the equation $\varepsilon \times dt \approx \Delta column \times \frac{\alpha}{\pi} \times \pi \times d \times v \times dt.$**





## 2.4 CH$_4$ background signal

The satellite data can be written as a vector **y**, where each element corresponds to an individual satellite data point. This signal
is caused by a CH$_4$ background signal and the CH$_4$ plume due to the emissions from the waste disposal sites near Madrid:

$$y = y_{BG} + y_{plume}$$ Eq. 7

It is of great importance to adequately separate both components for estimating the emission strength from the satellite data.

For determining the background signal ($y_{BG}$), we setup a background model:

$$M_{BG} = y_{BG} = K_{BG}x_{BG}$$ Eq. 8

The matrix $K_{BG}$ is a Jacobian matrix that allows to reconstruct the background according to a few background model
coefficients (the elements of the vector $x_{BG}$). We also create a Jacobian $K_{BG}^*$, which is the same as $K_{BG}$ but set to zero for
observations where the wind data suggest a significant impact of the CH$_4$ plume on the satellite data. The calculations of the
plume CH$_4$ signals are made according to Sect. 2.3. With the use of $K_{BG}^*$ we make sure that the estimated background signal
is not affected by the CH$_4$ plume.

The Jacobian matrix $K_{BG}$ considers a smooth background, which is a constant CH$_4$ value, a linear increase with time and a
seasonal cycle described by the amplitude and phase of the three frequencies 1/year, 2/year, and 3/year. Furthermore, we fit a
daily anomaly, which is the same for all data measured during a single day and a horizontal anomaly, which is the same for
any time but dependent on the horizontal location. For the latter we use a $0.1° \times 0.135°$ (latitude $\times$ longitude) grid.

We invert the problem in order to estimate the background model coefficients (elements of the vector $x_{BG}$):

$$\hat{x}_{BG} = G_{BG}y_{BG}$$ Eq. 9

With $G_{BG}$ being the so-called gain matrix

$$G_{BG} = (K_{BG}^{*T}S_{y,n}^{-1}K_{BG}^* + S_a^{-1})^{-1}K_{BG}^{*T}S_{y,n}^{-1}$$ Eq. 10

The matrix $S_{y,n}$ stands for the noise covariance of the satellite data. For constraining the problem, we use a diagonal $S_a^{-1}$
(no constraint between different coefficients) with a very low constraint value for the coefficient determining the constant and
higher constraint values for the other coefficients. Here we use $S_{y,BG}$ as the diagonal matrix with the mean square value of the
difference $y_{BG} - K_{BG}^*\hat{x}_{BG}$ being the diagonal elements. In this context, $S_{y,BG}$ considers the deficits of the background model.

The uncertainty of the background model coefficients can be calculated as:

$$S_{\hat{x}_{BG}} = G_{BG}S_{y,BG}G_{BG}^T$$ Eq. 11

For each day there is an uncertainty in the background coefficients and the uncertainty is correlated with the uncertainty at
other days. All this information is provided in the uncertainty covariance $S_{\hat{x}_{BG}}$.

With the full Jacobian $K_{BG}$ we can now model the background for the measurement state (also for the measurements that
are assumed to be affected by the CH$_4$ waste disposal plume):

$$y_{BG} = K\hat{x}_{BG}$$ Eq. 12

and calculate the plume signal according to Eq. 7 as:



$$y_{plume} = y - K\hat{x}_{BG} \qquad\qquad \text{Eq. 13}$$

The uncertainty of these plume signal is the sum of the uncertainties of the satellite data $S_{y,n}$ and the uncertainty of the

estimated background:

$$S_{y,plume} = S_{y,n} + KS_{\hat{x}_{BG}}K^T \qquad\qquad \text{Eq. 14}$$

## 2.5 Fitting of CH$_4$ emission rates

Because the CH$_4$ plume signal is rather weak compared to the CH$_4$ background uncertainty and the noise level of the satellite data, we have to work with averages in order to reduce the data noise. The averaging is made by classifying the observation in two predominant wind categories. We calculate the average plume maps for the southwest and northeast wind situations (see

Figure 6 and Figure 8). Then we calculate the difference between the south-west and north-east plume maps (the wind-assigned anomalies or Δ-maps). All the calculations are made by binning all observations that fall within a certain $0.135° \times 0.1°$ (longitude $\times$ latitude) area. In order to significantly reduce the data noise, we only consider averages for the $0.135° \times 0.1°$ areas based on at least 25 individual observations made under southwest wind conditions and 25 individual observations made under northeast wind conditions. The binning, the averaging, the wind-assigned Δ-maps calculations, and the data number filtering

is achieved by operator **D,** and we can write:

$$\Delta y_{plume} = Dy_{plume} \qquad\qquad \text{Eq. 15}$$

and

$$\Delta S_{y,plume} = DS_{y,plume}D^T \qquad\qquad \text{Eq. 16}$$

Here $\Delta y_{plume}$ is a column vector whose elements capture the different signal of the two wind directions at the different locations and $\Delta S_{y,plume}$ is the corresponding uncertainty covariance.

For modelling the plume signals we use a priori knowledge of CH$_4$ emission locations, i.e. assuming a repartition of the

emissions between the three waste disposal sites according to Table 3 (see Sect. 3.1). Together with information from the wind, we then model the CH$_4$ plume's wind-assigned anomaly signal $\Delta y_{plume}$:

$$\Delta y_{plume} = \Delta kx \qquad\qquad \text{Eq. 17}$$

Here the Jacobian $\Delta k$ (a column vector) represents the wind-assigned anomaly model as described in Sect. 2.3. It describes how an emission at the waste disposal sites according to Table 3 would be seen in the difference signal. We are interested in the coefficient $x$ (a scalar describing how the assumed emissions from Table 3 have to be scaled by a common factor in order

to achieve the best agreement with the observed plume).

Similar to Eq. 9 and Eq. 10 we write:

$$\hat{x} = g^T \Delta y_{plume} \qquad\qquad \text{Eq. 18}$$

with the row vector

$$g^T = \left(\Delta k^T \Delta S_{y,plume}^{-1} \Delta k\right)^{-1} \Delta k^T \Delta S_{y,plume}^{-1} \qquad\qquad \text{Eq. 19}$$



This fitting of the emission rate correctly considers the respective uncertainty of the difference signals at the different locations.

Because of the small plume signals, it is important to estimate the reliability of the fitted emission rate. The uncertainty of

$x$ due to the background uncertainty and the noise in the satellite data can be estimated as:

$$\epsilon_{BG} = \sqrt{\boldsymbol{g}^T \mathbf{DKS}_{\hat{x},\mathbf{BG}} \boldsymbol{K}^T \mathbf{D^T} \boldsymbol{g}}$$ 

**Eq. 20**

and

$$\epsilon_n = \sqrt{\boldsymbol{g}^T \mathbf{DS_{y,n}} \mathbf{D^T} \boldsymbol{g}}$$ 

**Eq. 21**

respectively.

## 3 Results and discussion

### 3.1 Validation of TROPOMI using COCCON measurements

Figure 3 shows the correlation between COCCON and TROPOMI measurements. The here shown TROPOMI data are the mean value of observations collected within a radius of 5 km around each COCCON station. The coincident COCCON $XCH_4$ is the mean value of the measurements within 30 minutes before or after TROPOMI overpass. The distance between two stations ranges between 6 km and 14.2 km. The TROPOMI data within a circle with a large radius might cover the information from other stations nearby, which brings error in the correlation between the coincident data. Therefore, we choose 5 km as

the radius of collection circle for TROPOMI and the coincident data at each station show generally good agreement. It shows that TROPOMI data have good quality and low bias. Note that there are 1 to 2 TROPOMI measurements located within a circle of 5 km radius around each station.

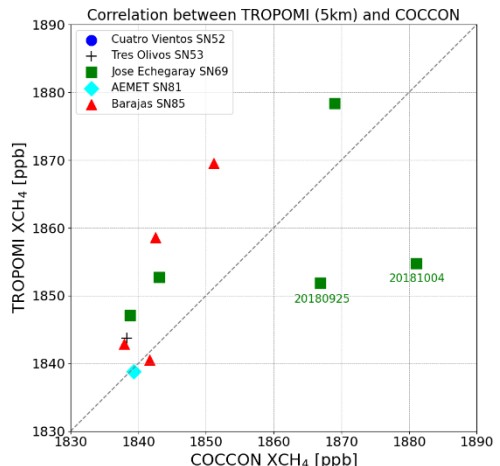

**Figure 3: Correlation plot between TROPOMI observations collected within 5 km radius around each COCCON station and**
**coincident COCCON (within 30 minutes) measurements at five stations in 2018.**





The coincident data on September 25, 2018 and October 4, 2018 show large biases at Jose Echegaray station where the SN69 COCCON instrument is located. Due to its coarser spatial resolution the TROPOMI $XCH_4$ observations do not capture the local enhancements detected by the COCCON instrument in the vicinity of the assumed source. Figure 4 illustrates the two exemplary days of the time series of COCCON SN69 and coincident TROPOMI observations. Obvious enhancements are

observed at around 13:00 UTC on September 25 and at around 12:30 on October 4, 2018 (see Figure A- 2 for the other days). It is noted that the $XCH_4$ enhancements can also be observed by the other stations when the $CH_4$ plume passes over Madrid. We only discuss the two representative days here, as we focus on the specific source near the Jose Echegaray station. The Valdemingómez and Pinto waste plants are located nearby with a distance of 4.5 km and 12 km, respectively.

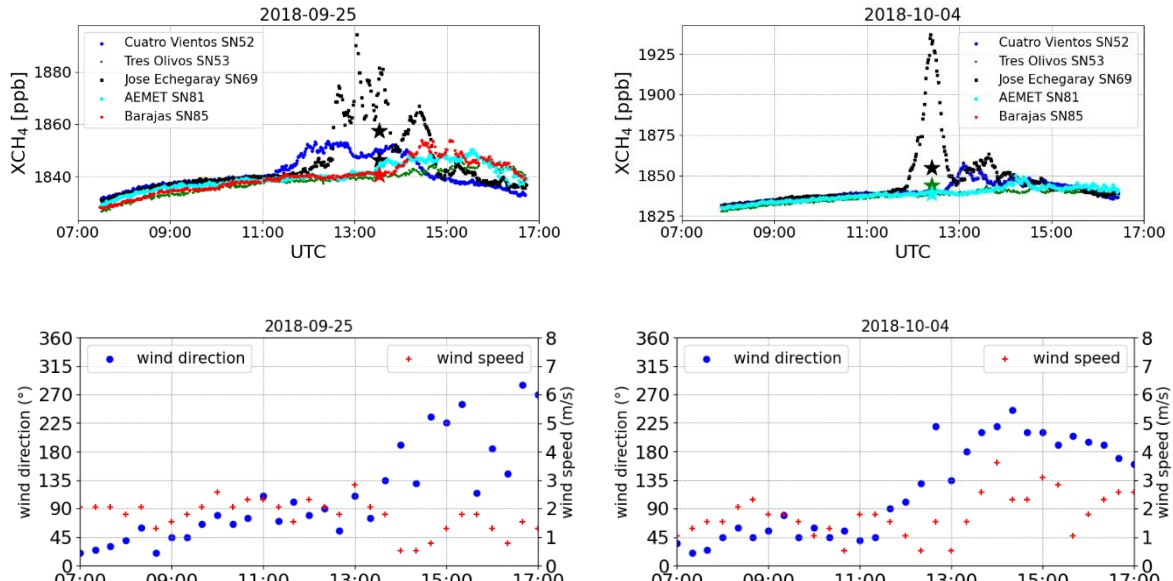

**Figure 4: Time series of COCCON measurements at five stations on two days in 2018. Star symbols represent the averaged TROPOMI observations within a radius of 5 km around each station. Lower panels show the wind direction and wind speed measured at the Cuatro Vientos Airport.**

TROPOMI detected 10 ppb higher $XCH_4$ at Jose Echegaray station than at Barajas station on September 25, 2018. However,

COCCON observed much higher amount of $XCH_4$ (53 ppb) at Jose Echegaray station than at Barajas station (and other stations) at around 13:00 UTC. The delayed enhancement at AEMet and Barajas stations at the downwind direction is found after the wind direction changed from north more towards south direction. Another obvious enhancement of $XCH_4$ is observed at Jose Echegaray station by the COCCON SN69 instrument at around 12:30 on October 4, 2018, with about 97 ppb higher $XCH_4$ than COCCON measurements at the other four stations. However, TROPOMI only measured about 13 ppb higher $XCH_4$ at

Jose Echegaray station compared to the TROPOMI measurements at the other stations. These considerable enhancements at Jose Echegaray station observed by the COCCON instrument are likely due to the local source (the nearby Valdemingómez waste plant). The plume is in close vicinity to the source narrower than the pixel scale of the satellite, and therefore is only detected as attenuated signal by TROPOMI. The full width at the half maximum (FWHM) of the enhancement peak on October





4, 2018 roughly covers a time period of 30min, with a corresponding wind direction change of 22.5° (~0.4 rad) and an averaged
wind speed of 1.0 m s⁻¹. The distance between the COCCON SN69 to the Valdemingómez waste plant is about 4500 m. Then
the 97 ppb enhancement measured by COCCON SN69 instrument yields an estimated emission strength of $3.7 \times 10^{25}$ molec s⁻¹.

   According to the Spanish Register of Emissions and Pollutant Sources (PRTR, http://www.en.prtr-es.es/, last access: 20
February, 2021), more than 95% of total $CH_4$ emissions are from three waste treatment and disposal plants in the Madrid
region (locations showed in Figure 1). The annual $CH_4$ emission rates from the PRTR for each plant are listed in Table 3. The
total emission strength for each plant is about $2.5 \times 10^{25}$ molec s⁻¹. This value only considers the "cells" in production, i.e. those
where the waste is not yet covered with soil. The emissions from sealed cells are not included in the total emissions, but they
still emit $CH_4$ during some years after sealing. So, the estimated emission rates from inventories is expected to underestimate
the true emissions, which fits reasonably with the estimated emission rate derived from COCCON measurements. The
COCCON instruments show a very good ability to detect the source. Based on this evidence we investigate the potential of
the TROPOMI and IASI $CH_4$ products for detecting $CH_4$ sources later in the following.

**Table 3: CH₄ emission rates in three waste treatment and disposal plants in Madrid from PRTR.**

| Waste treatment and disposal plants | Valdemingómez (molec s⁻¹) | Pinto (molec s⁻¹) | Alcalá (molec s⁻¹) | Total (molec s⁻¹) |
|---|---|---|---|---|
| 2017 | $7.4 \times 10^{24}$ | $1.2 \times 10^{25}$ | $2.1 \times 10^{24}$ | $2.2 \times 10^{25}$ |
| 2018 | $7.4 \times 10^{24}$ | $1.3 \times 10^{25}$ | $2.1 \times 10^{24}$ | $2.2 \times 10^{25}$ |
| 2019 | $9.8 \times 10^{24}$ | $1.4 \times 10^{25}$ | $9.4 \times 10^{23}$ | $2.5 \times 10^{25}$ |

**3.2 Predominant wind**

To better representing the whole area of Madrid, the hourly ERA5 model wind at a height of 10m around Madrid is used.
ERA5 is the fifth generation climate reanalysis produced by the European Centre for Medium-range Weather Forecasts
(ECMWF) (Copernicus Climate Change Service, 2017). The TROPOMI overpasses over Madrid cover the time range from
12:00 UTC – 14:30 UTC (IASI overpasses are typically from 09:30 UTC – 10:30 UTC), but the dispersion of emitted $CH_4$ is
influenced by the ground conditions (e.g. wind speed and wind direction) over a wider time range (Delkash et al., 2016; Rachor
et al., 2013). Therefore, the wind information between daytime (08:00 UTC – 18:00 UTC) is chosen to define the predominant
wind direction for each day. Figure 5 presents the wind roses for daytime between 10 November 2017 and 10 October 2020
(the first and last day with valid TROPOMI data). The dominating wind direction was southwesterly. To the northwest of
Madrid are the Guadarrama mountains located and the Jarama and Manzanares river basins, which influence the air flow.
Therefore, we use a wider wind range for specific wind area in this study to cover the dominant wind directions, i.e. SW for
the range of 135° – 315° and NE for the remaining direction. If a wind direction dominates 60% of records for one day, i.e., if
the wind direction belongs to one specific area more than 60% of the daytime (08:00 UTC – 19:00 UTC), then this predominant





wind direction is selected for that day. The SW and NE wind fields are used for constructing wind-assigned anomalies in this study and we will demonstrate this construction by using TROPOMI nitrogen dioxide ($NO_2$) data in the next section. Table 4 summaries the number of days and wind speed for each specific wind area. The wind direction during the TROPOMI overpasses was 61.8% in SW wind field and 28.4% in NE wind field and their averaged wind speed is similar.

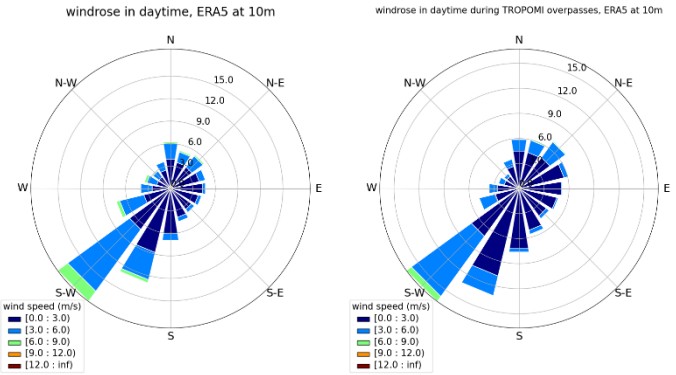


**Figure 5: Wind roses for daytime (08:00 UTC – 19:00 UTC) from 10 November 2017 to 10 October 2020 for the ERA5 model wind. The left panel covers all days and the right panel covers the days with TROPOMI overpasses.**

**Table 4: Number of days and the averaged ERA5 wind speed (± standard deviation) per specific wind area in daytime (08:00 UTC – 18:00 UTC) from 10 November 2017 to 10 October 2020. Columns 2 and 3 are for all days, and columns 4 and 5 are for days with**

**TROPOMI overpass.**

|  |  |  | **TROPOMI overpass** | |
| --- | --- | --- | --- | --- |
| **Wind direction range** | **Number of days in total (%)** | **Averaged wind speed ± standard deviation (m s⁻¹)** | **Number of days in total (%)** | **Averaged wind speed ± standard deviation (m s⁻¹)** |
| NE / >315° or <135° | 30.4 | 2.6 ± 1.5 | 28.4 | 2.3 ± 1.2 |
| SW / 135° – 315° | 68.4 | 2.8 ± 1.7 | 61.8 | 2.3 ± 1.4 |

### 3.3 Illustration and validation of the wind-assigned anomaly method

When fossil fuels are burned, nitrogen monoxide (NO) is formed and emitted into the atmosphere. NO reacts with $O_2$ to form $NO_2$ and with ozone ($O_3$) to produce $O_2$ and $NO_2$. $NO_2$ is an extremely reactive gas with a short lifetime of a couple of hours and has lower background levels than $CH_4$. It is measured by TROPOMI with excellent quality. Therefore, it is a suitable

proxy for demonstrating the method developed for the wind-assigned anomaly.

TROPOMI offers simultaneous observations of $NO_2$ columns with a recommended quality value for the analysis of TROPOMI $NO_2$ columns of qa>0.75 (http://www.tropomi.eu/sites/default/files/files/publicSentinel-5P-Level-2-Product-User-Manual-Nitrogen-Dioxide.pdf). Based on the predominant wind direction in Madrid (see section 3.2), the averaged wind-assigned anomalies are defined here as the difference of the mean TROPOMI $NO_2$ column under the wind direction from NE

and the mean TROPOMI $NO_2$ column under the predominant wind direction of SW in Madrid.

Figure 6 (a) illustrates the wind-assigned anomalies of TROPOMI $NO_2$ ($\Delta NO_2$) on a 0.1° × 0.135° latitude/longitude grid during 2018 – 2019. Pronounced fusiform-shape plumes are observed along NE – SW wind direction as expected. Figure 6





(b) shows the wind-assigned anomalies derived from the simple model introduced in Sect. 2.3, using Madrid city center as the point source and an assumed emission rate ($\varepsilon$) of $5.0\times10^{24}$ molec s$^{-1}$ and using ERA5 10 m wind data. The similar symmetrical

positive and negative plumes to those in Figure 6 (a) imply that our method of wind-assigned anomaly is working nicely, and that the ERA5 10 m data are indeed representative for the area and that the implementation of the satellite data analysis is correct. Figure 6 (c) shows the strong correlation between the wind-assigned anomalies derived from the TROPOMI measurements and the simple plume model ($\varepsilon = 5.0\times10^{24}$ molec s$^{-1}$). Using the fitting method as described in Sect. 2.5, we estimate an emission rate of $3.5\times10^{24}$ molec s$^{-1} \pm 3.9\times10^{22}$ molec s$^{-1}$. Here the uncertainty is due to noise of the observations

and is calculated according to Eq. 21. This estimated source strength is weaker than the strength obtained by Beirle et al. (2011), where the reported NO$_x$ emission is around 150 mol s$^{-1}$ in Madrid, corresponding to a NO$_2$ emission of $6.8 \times10^{25}$ molec s$^{-1}$. It is because our model does not consider the decay of NO$_2$, which results in a lower emission rate.

The result of this test using NO$_2$ also allows the angular spread parameter in the plume model used to be adjusted (see Section 2.3 and equation (3)). As it can be seen from Figure A-1, assuming an angular spread of 60° reasonably reproduces

the shape of the plume.

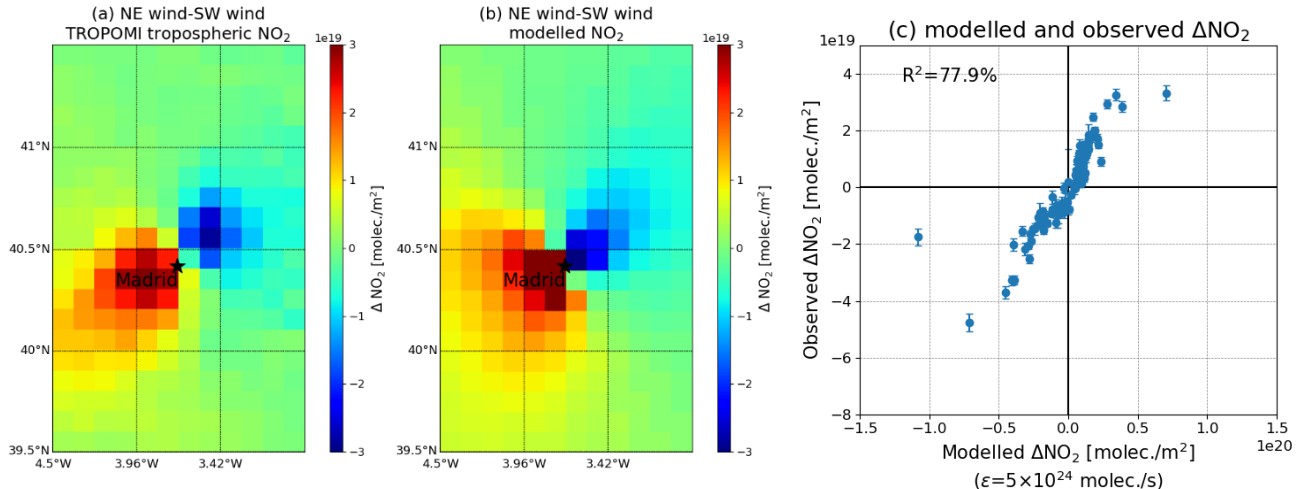

**Figure 6 Wind-assigned anomalies derived from (a) TROPOMI tropospheric NO₂ column, derived from (b) our simple model (ε= 5×10²⁴ molec s⁻¹) over Madrid in NE - SW direction on a 0.1° × 0.135° latitude/longitude grid during 2018 – 2020, and (c) the correlation plot between observed ΔNO₂ and modelled ΔNO₂ (ε=5×10²⁴ molec s⁻¹) during 2018-2019.**

### 3.4 XCH₄ and TXCH₄ anomaly

CH₄ has a relatively longer lifetime than NO$_2$ and its background in the atmosphere is high. An increasing trend with obvious seasonality and strong day-to-day signals for XCH$_4$ are seen in Figure 7 (upper panels). Therefore, these background signals need to be removed in order to reveal before simulating the wind-assigned anomalies (see Sect. 2.4). After removing the background, the anomalies (raw data - background) represent more or less the emission from local area (Figure 7 lower panels).





Figure 8 illustrates the anomalies of XCH$_4$, TXCH$_4$ and UTSXCH$_4$ for all measurement days, days under SW wind field and days under NE wind field. The distributions over the whole area for XCH$_4$ and TXCH$_4$ are similar and no obvious enhancement is observed in UTSXCH$_4$, as CH$_4$ abundances dominate in the troposphere. The areas where the three waste plants locate show obvious high anomalies in downwind direction, demonstrating that our method of removing background works well and the satellite products can detect the local pollution sources after removing the background. Enhanced plumes
of XCH$_4$ and TXCH$_4$ are better visible in the downwind side of SW than in the downwind side of NE wind field. Because SW is the most dominant wind direction, the SW plume signal is based on a higher number of data and thus less noise.










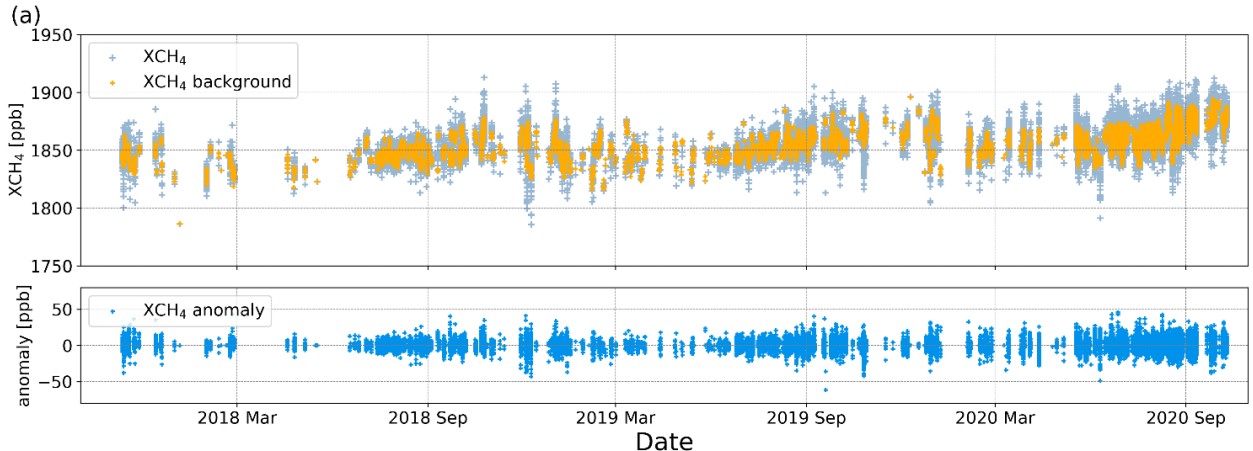

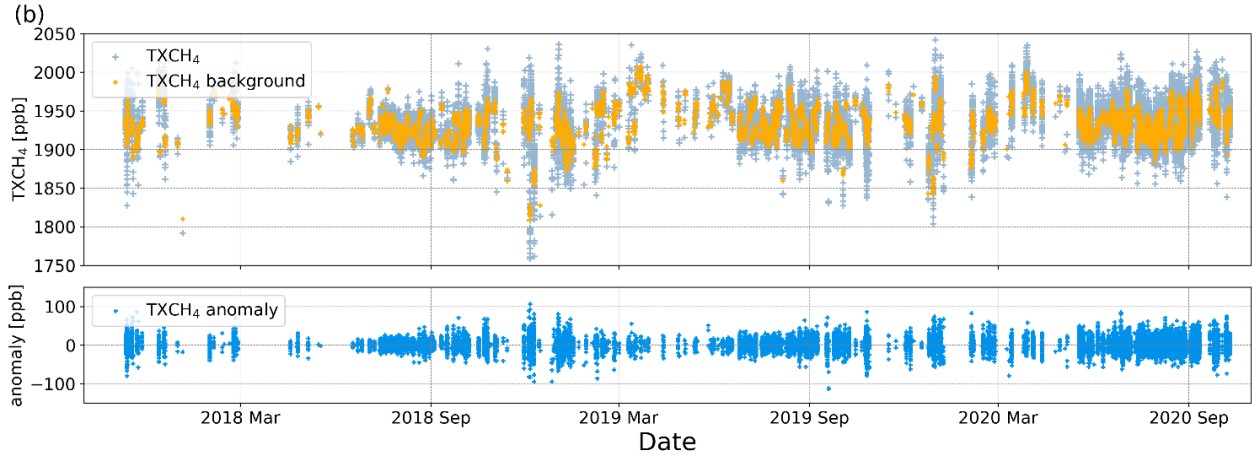

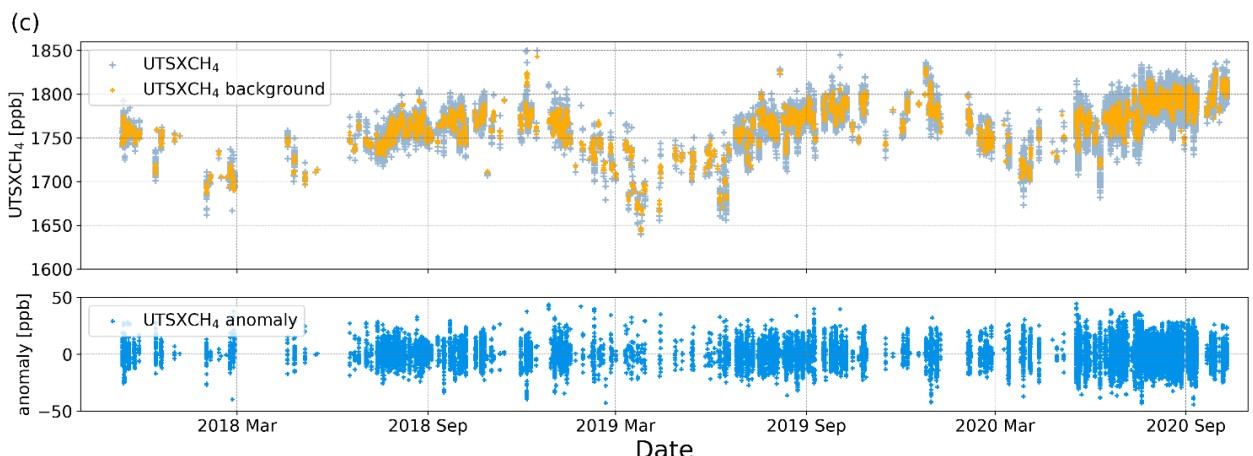

**Figure 7 Time series of (a) XCH4, (b) TXCH4 and (c) UTS XCH4, showing raw data and background in each upper panel and anomalies in each corresponding lower panel.**



**Figure 8 (a-c) XCH₄, (d-f) TXCH₄ and (g-i) UTSXCH₄ anomalies for all days, days with SW wind and NE wind directions. The triangle symbols represent the location of waste plants.**





### 3.5 Estimation of CH$_4$ emission strengths from satellite data sets

The wind-assigned anomalies derived from XCH$_4$ anomalies and TXCH$_4$ anomalies on a 0.1° × 0.135° latitude/longitude grid are presented in Figure 9. The XCH$_4$ and TXCH$_4$ wind-assigned anomalies show similar bipolar plumes but more disturbed compared to those in NO$_2$. This is because the CH$_4$ signal is weak compared to the background concentration, so the noise level of the measurement and the imperfect elimination of the background are significant disturbing factors.

Based on the a priori knowledge of the locations of the three waste plants, we choose their locations as point sources to model the enhanced XCH$_4$ according to the wind information. The initial emission strength is 1×10$^{26}$ molec s$^{-1}$ in total and the emission rate at each point source is repartitioned among these three sites according to Table 3. The modelled and observed wind-assigned anomalies show a reasonable linear correlation (coefficient of determination R$^2$ of about 49% and 44% for XCH$_4$ and TXCH$_4$, respectively) with observed ΔXCH$_4$. Based on Eq. 18, we obtained an estimated emission rate of 7.4×10$^{25}$ ± 6.4×10$^{24}$ molec s$^{-1}$ for XCH$_4$ and 7.1×10$^{25}$ ± 1.0×10$^{25}$ molec s$^{-1}$ for TXCH$_4$. The uncertainty values given here are the square root sum of the uncertainty due to the background signal and the data noise, which are calculated according to Eq. 20 and 21. Figure 9 (g), (h) and (i) show the wind-assigned anomalies for UTSXCH$_4$. For the modelled UTSXCH$_4$ anomalies we assume here the CH$_4$ enhancement to occur at altitudes between 7 and 20 km a.s.l. As expected, the fit of these model data to the observed UTSXCH$_4$ data yields emission rates of close to zero (1.4×10$^{25}$ ± 7.2×10$^{24}$ molec s$^{-1}$), revealing that there is no significant plume signal above 7 km a.s.l. The fact that for TXCH$_4$ we obtain practically the same emission rates as for XCH$_4$ and that in the UTSXCH$_4$ data we see almost no plume nicely proves the quality of our careful background treatment method and the low level of cross-sensitivity between the TXCH$_4$ and UTSXCH$_4$ data products. The applied background treatment allows detecting the surface-near emission signal consistently in the total column XCH$_4$ data and in the tropospheric TXCH$_4$ data.

Figure 10 illustrates the estimated emission strengths for the different products. The emission strengths derived from the satellites are higher than that derived from COCCON measurements, as TROPOMI covers larger area and COCCON only measured local sources. Table 5 presents some estimated emissions for CH$_4$ from other studies. Our results fit well with the Indianapolis emission and is reasonably comparable to the emission from another European metropolitan city of similar characteristics, like Berlin.

**Figure 9: Wind-assigned XCH₄ plume derived from (a) TROPOMI XCH₄ anomalies, (d) synergetic TXCH₄ anomalies and (g) UTSXCH₄ anomalies and their corresponding modeled plume (b, e, h) over Madrid in NE – SW direction on a 0.1° × 0.135° latitude/longitude grid. The correlation plots between observed ΔXCH₄ and modelled ΔXCH₄ (ε=1×10²⁶ molec s⁻¹) for different products (c, f, i). Here we use the three waste plants as the point sources (blue triangle with red edge color). The initial emission rate in model is 1×10²⁶ molec s⁻¹ and proportionally distributed into three point sources based on the a priori knowledge of emission rate in each waste plant. For the modelled UTSXCH₄ anomalies we assume the CH₄ enhancements to occur at altitudes between 7 and 20 km a.s.l.**





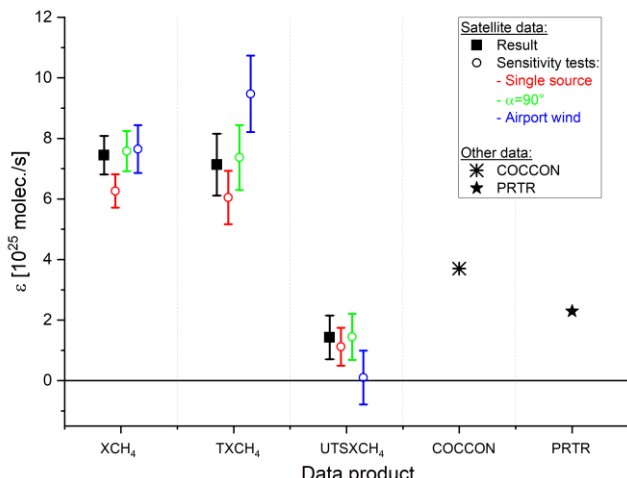

**Figure 10 Emission strengths for different products with sensitivity tests. Also included are the COCCON observations which characterizes the Valdemingómez waste plant contribution and the total of all three sources according to the PRTR inventory.**

470 **Table 5: CH$_4$ estimated emissions in some studies.**

| Reference | Location (specific source) | Estimated emissions (molec s$^{-1}$) |
|---|---|---|
| Cambaliza et al., 2015 | Indianapolis | $8.1 \times 10^{25}$ |
| Luther et al., 2019 | Upper (coal mining) | $7.1 \times 10^{24} - 1.3 \times 10^{26}$ |
| Klausner et al., 2020 | Berlin | $2.0 \times 10^{26}$ |
| Alvarez et al., 2018 | U.S. (oil and natural gas supply chain) | $1.6 \times 10^{28}$ |

**3.6 Sensitivity study for emission strength estimates**

The point sources and their proportion in the total emission rate in this study are based on the a priori knowledge of three different waste plant locations. If we use a single source located at the Pinto waste disposal site only, it yields an emission rate of $6.3 \times 10^{25}$ molec s$^{-1}$, ~15% lower than that of the three-point sources for XCH$_4$ and $6.0 \times 10^{25}$ molec s$^{-1}$ (-15%) for TXCH$_4$

475 (see Figure 10). The opening angle ($\alpha$) is experimentally selected based on the comparison between the TROPOMI measured and modelled NO$_2$ plume, which results in some uncertainties as well. Using 90° instead of 60° for $\alpha$ in the plume model results in an emission strength of $7.6 \times 10^{25}$ molec s$^{-1}$ (+3% change) for XCH$_4$ and of $7.4 \times 10^{25}$ molec s$^{-1}$ (+4% change) for TXCH$_4$.

The surface wind can be influenced by the topography and the actual transport pathway from emission source to the

480 measurement station is difficult to know (Chen et al., 2016; Babenhauserheide et al., 2020). To study the wind sensitivity, the hourly wind information measured at the Cuatro Vientos Airport at 10m height is used instead of the ERA5 10 m wind. There are other in situ measurements available but not used here, as the AEMet Headquarter station is affected by nearby buildings and the Barajas Airport station is very close to a secondary river (Jarama) that determines a specific wind pattern. The wind





measured at the Cuatro Vientos Airport is quite different compared to the ERA5 wind, as in situ measured NE wind becomes
dominant as well and the wind speed in SW wind field increases by ~50% compared to that of ERA5 wind (Figure A- 3, Figure
A- 4 and Table A- 1). Using the wind measured at the Cuatro Vientos Airport results in an emission rate of $7.7 \times 10^{25}$ molec s$^{-1}$ (+4%) for XCH$_4$ and $9.5 \times 10^{25}$ molec s$^{-1}$ (+34%) for TXCH$_4$.

In summary, the uncertainties derived from source location, opening angle or wind cannot be ignored, but nevertheless the
emission rates estimated from the space-borne observations are clearly larger than the values reported in Table 3 and larger
than that estimated from the COCCON SN69 observations in October 2018.

## 4 Conclusions

The present study analyzes TROPOMI XCH$_4$ and IASI CH$_4$ retrievals over an area around Madrid for more than 400 days
within a rectangle of 39.5°N – 41.5°N and 4.5°W – 3.0°W (125 km × 220 km) from 10 November 2017 until 10 October 2020.
During this time period, a two-week field campaign was conducted in September 2018 in Madrid, in which five ground-based
COCCON instruments were used to measure XCH$_4$ at different locations around the city center.

First, TROPOMI XCH$_4$ is compared with co-located COCCON data from the field campaign, showing a generally good
agreement, even though the radius of the collection circle for the satellite measurements is as low as 5 km. However, there are
six days when obvious enhancements due to local sources were observed by COCCON around noon at the most southeast
station (Jose Echegaray), which were underestimated by TROPOMI. The ground-based COCCON observations indicate a
local source strength of $3.7 \times 10^{25}$ molec s$^{-1}$ from observations at Jose Echegaray station on October 4, 2018, which is reasonable
compared to the emissions assumed for nearby waste plants. The waste plant locations are later used as the point sources to
model the emission strength for CH$_4$.

According to ERA5 model wind at 10 m height, SW (135° – 315°) winds (NE covering the remaining wind field) are
dominant in the Madrid city center in the time range from November 2017 to October 2020. Based on this wind information,
the wind-assigned anomalies are defined as the difference of satellite data between the conditions of NE wind field and SW
wind field. We use simultaneous tropospheric NO$_2$ column amount from TROPOMI as a proxy to evaluate the wind-assigned
anomaly approach due to its short lifetime, by using ERA5 model wind. Pronounced and bipolar NO$_2$ plumes are observed
along NE – SW wind direction and center of Madrid city and an estimated tropospheric NO$_2$ emission strength of $3.5 \times 10^{24}$ ±
$3.9 \times 10^{22}$ molec s$^{-1}$ is calculated. This implies that our method of wind-assigned anomaly is working reliably, and that the
ERA5 wind data used are indeed representative for the area and the implementation of the satellite data analysis is correct.

CH$_4$ has a long lifetime and so there are strong CH$_4$ background signals in the atmosphere. Therefore, this background
values are needed to be removed and the anomalies have to be determined before calculating emission strengths. In this study,
the background values include linear increase, seasonal cycle, daily variability and horizontal variability. The areas where the
three waste plants locate show obvious high anomalies, demonstrating that the satellite products can detect the local sources





after removing the background. Enhanced plumes are pronounced in the downwind side of SW, whereas the observed downwind plume signal for NE wind is noisier, partly due to the lower number of NE wind situations.

The wind-assigned TROPOMI $XCH_4$ anomalies show a less clear bipolar plume than $NO_2$. This is because $CH_4$ has a long lifetime and its high background is difficult to be totally removed. Based on the wind-assigned method, the emission strength estimated from the TROPOMI $XCH_4$ data is $7.4\times10^{25} \pm 6.4\times10^{24}$ molec s$^{-1}$. In addition, this method is applied to the

tropospheric partial column averaged (ground – 7 km a.s.l.) dry-air molar fractions of methane ($TXCH_4$, obtained by combing TROPOMI and IASI products) yielding an emission strength of $7.1\times10^{25} \pm 1.0\times10^{25}$ molec s$^{-1}$. We show that in the upper troposphere/stratosphere there is no significant plume signal ($1.4\times10^{25} \pm 7.2\times10^{24}$ molec s$^{-1}$). The estimation of very similar emission rates from $XCH_4$ and $TXCH_4$ together with the estimated negligible emission rates when using data representing the upper troposphere/stratosphere proves the robustness of our method. The emission rates derived from satellites ($XCH_4$ and

$TXCH_4$) are higher than that derived from COCCON observations, as satellites cover larger areas with other $CH_4$ sources and COCCON likely measures local sources.

The surface wind is easily influenced by the topography, which introduce uncertainties in the estimated emission strengths. Using in situ measured wind at the Cuatro Vientos Airport instead of ERA5 model wind results in an estimated emission rate of $7.7\times10^{25}$ molec s$^{-1}$ (+4%) for $XCH_4$ and $9.5\times10^{25}$ molec s$^{-1}$ (+34%) for $TXCH_4$. Uncertainties can be caused by the choice

of opening angle in plume model as well. The estimated emission rates with α=90° are $7.6\times10^{25}$ molec s$^{-1}$ (+3%) for $XCH_4$ and of $7.4\times10^{25}$ molec s$^{-1}$ (+4%) for $TXCH_4$. When using single source located in the Madrid city center, the emission strengths are $6.3\times10^{25}$ molec s$^{-1}$ (-15%) for $XCH_4$ and $6.0\times10^{25}$ molec s$^{-1}$ (-15%) for $TXCH_4$.

In summary, in this study for the first time TROPOMI observations are used together with IASI data and the ground-based COCCON observations to investigate $CH_4$ emissions from an important metropolitan area like the Madrid city. The COCCON

instruments show a promising potential for satellite validation and an excellent ability for observation of local sources. The data presented here are provided with the confidence that TROPOMI is able to detect the tropospheric $NO_2$ and $XCH_4$ anomalies over metropolitan areas with support from meteorological wind analysis data. As outlook, this methodology could also be applied to other source regions, space-based sensors and sources of $CO_2$.





*Data availability.* The data are accessible by contacting the corresponding author (qiansi.tu@kit.edu). The SRON S5P-RemoTeC scientific TROPOMI CH4 dataset from this study is available for download at

https://doi.org/10.5281/zenodo.4447228 (Lorente et al., 2021, last access: 06 May 2021). The TROPOMI data set is publicly available from https://scihub.copernicus.eu/ (last access: 06 May 2021; ESA, 2020). The access and use of any Copernicus Sentinel data available through the Copernicus Open Access Hub are governed by the legal notice on the use of Copernicus Sentinel Data and Service Information, which is given here: https://sentinels.copernicus.eu/documents/247904/690755/Sentinel_Data_Legal_Notice (last access: 06 May 2021; European

Commission, 2020). The MUSICA IASI data set is available for download via https://doi.org/10.35097/408 (Schneider et al. 2021c).

*Author contributions.* Qiansi Tu, Frank Hase and Omaira García developed the research question. Qiansi Tu wrote the manuscript and performed the data analysis with input from Frank Hase, Omaira García, Matthias Schneider and Farahnaz

Khosrawi. Frank Hase suggested the method of constructing wind-assigned anomalies for source quantification. Matthias Schneider suggested the method for calculating the anomalies, for fitting the emission rates and estimating the uncertainty. Omaira García provided the COCCON and meteorological data and helped to interpret it. Tobias Borsdorff and Alba Lorente supported technically for the TROPOMI data analysis. Matthias Schneider, Benjamin Ertl and Christopher J. Diekmann provided the combined (MUSICA IASI + TROPOMI) data and supported technically for the analysis of these data. All other

coauthors participated in the field campaign and provided the data. All authors discussed the results and contributed to the final manuscript.

*Competing interests.* The authors declare that they have no conflict of interest.

*Acknowledgements.* We acknowledge ESA support through the COCCON-PROCEEDS and COCCON-PROCEEDS II projects. In addition, this research was funded by the Ministerio de Economía y Competitividad from Spain through the INMENSE project (CGL2016-80688-P). This research has largely benefit from funds of the Deutsche Forschungsgemeinschaft (provided for the two projects MOTIV and TEDDY with IDs/290612604 and 416767181, respectively). Important part of this work was performed on the supercomputer ForHLR funded by the Ministry of Science, Research and the Arts Baden-

Württemberg and by the German Federal Ministry of Education and Research.

We acknowledge the support by the Deutsche Forschungsgemeinschaft and the Open Access Publishing Fund of the Karlsruhe Institute of Technology.



**Appendix A**



**Figure A- 1 Examples of wind-assigned NO₂ plume based on the simple plume model ($\varepsilon = 5.0 \times 10^{24}$ molec s$^{-1}$) using Madrid as the point source in NE – SW direction on a 0.1° × 0.135° latitude/longitude grid with different opening angle (α) from 10° to 90°.**







**Figure A- 2 Time series of COCCON measurements at five stations and corresponding time series of wind fields (direction and speed) measured at the Cuatro Vientos Airport on eight days during MEGEI-MAD campaign in 2018. Star symbols represent the**
**TROPOMI observations within a radius of 5 km around each station.**





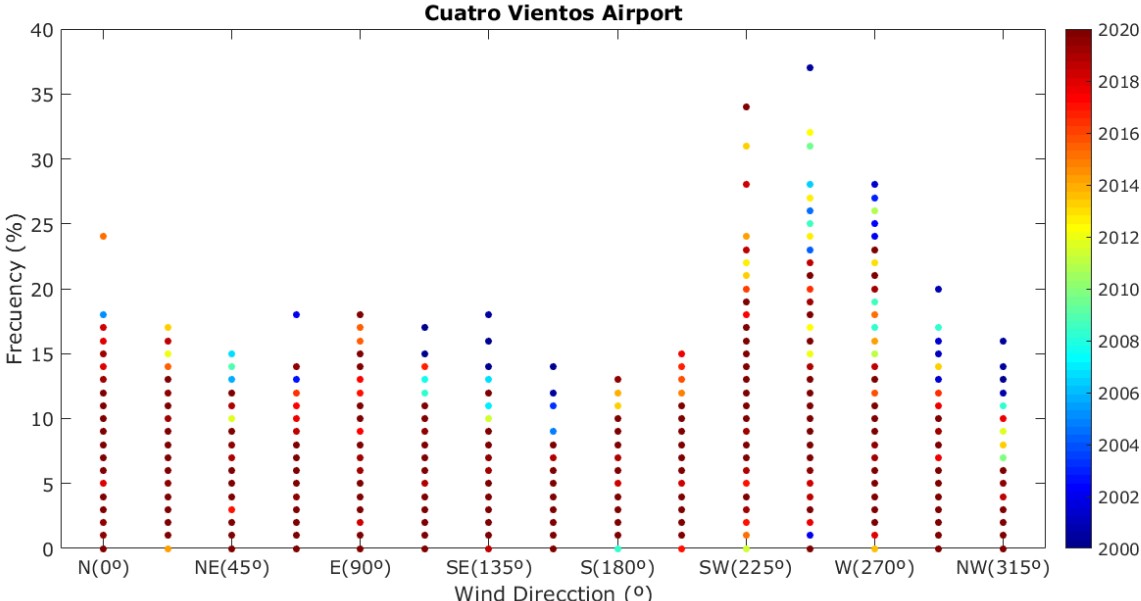

**Figure A- 3 Percentage of occurrence for wind direction measured at the Cuatro Vientos Airport between 2000 and 2020. The predominant wind direction is southwest and up to 35% of time (personal communication of Omaira García).**

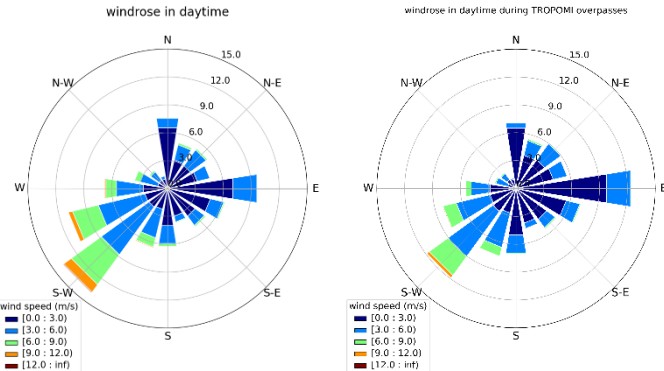

**Figure A- 4 Wind roses for daytime (08:00 UTC – 19:00 UTC) from 10 November 2017 to 11 September 2020 from the wind measurements at the Cuatro Vientos Airport. The left panel covers all days and the right panel covers the days with TROPOMI overpasses.**

**Table A- 1 Number of days and the averaged wind speed (± standard deviation) per specific wind area in daytime (08:00 UTC – 18:00 UTC) from 10 November 2017 to 11 September 2020 measured at the Cuatro Vientos Airport. Columns 2 and 3 are for all days, and columns 4 and 5 are for days with TROPOMI overpass.**

| Wind direction range | Number of days in total (%) | Averaged wind speed ± standard deviation (m s$^{-1}$) | TROPOMI overpass | |
| --- | --- | --- | --- | --- |
| | | | Number of days in total (%) | Averaged wind speed ± standard deviation (m s$^{-1}$) |
| NE / >315° or <135° | 35.4 | 2.4 ± 1.5 | 36.0 | 2.2 ± 1.3 |
| SW / 135° − 315° | 49.3 | 4.2 ± 2.5 | 44.4 | 3.4 ± 2.1 |



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
