# Peer review of "Quantification of CH4 emissions from waste disposal sites near the city of Madrid using ground- and space-based observations of COCCON, TROPOMI and IASI"

_Atmospheric Chemistry and Physics, 2021_

## Referee Report (RR1)

Reviewer - ACP Tu et al., 2021

**General:**

The authors apply both the ground-based (five COCCON instruments) and space-based (TROPOMI and TROPOMI+IASI) measurements to calculate the landfill CH4 emissions in the Madrid area. This study is practically interesting for the policy-makers in Spain, and it is also interesting for the relative scientist who would like to use these kinds of measurements to calculate the CH4 emissions at other places. In general, the topic fits well in the journal, the method is reasonable, and the uncertainties are considered. However, as the Referee #1 pointed out, there are some English grammar mistakes in the current version. In addition, the introduction lacks useful information and the formula remains unclear. I would like to recommend it to publish on ACP after answering my following comments.

**Major comments**

- Reconsider the structure of the introduction. For the moment, I cannot capture what is the current status of the CH4 emission research from the ground-based and spacebased measurements, what are the key issues, and what the authors will do in this work to solve/improve the issues. For example, I would like to suggest moving some texts in Section 2.1 and Section 2.2 to the introduction. Only keep the data description/technical part in Section 2.
- 2. In the abstract: "As CH4 emission strength we estimate 7.4×1025±6.4×1024 molec s-1 from the TROPOMI XCH4 data and 7.1×1025±1.0×1025 molec s-1 from the TROPOMI+IASI merged TXCH4 data." Why the uncertainty derived from the TROPOMI+IASI TXCH4 data is larger than that from the TROPOMI data? I thought that the advantage of using TROPOMI+IASI is to obtain more information in the troposphere so that users can reduce the uncertainty.

**Minor/technical comments:**

P2line36. It is confused with "that this". Please reword this sentence" That this strength is lower than the one derived from the satellite observations is a plausible result. ".

P2line45: "are to" -> "are"

P2line49: 'while' -> 'in which'

P2line49: What does '~55% uncertainty' mean? Please clarify it in the text.

P2line53: 'space borne' -> 'space-borne'. As you use both ground-based and space-based measurements in the study, why do you only highlight the space-based data here?

P2line62: 'TCCON' for the first time, please write down the full name

P3line70: 'column-average' -> 'column-averaged'

P3line81: 'The BrukerEM27/SUN ' -> 'A BrukerEM27/SUN'

- P4line115: 'we apply a strict quality control' -> 'we apply strict quality control'
- P4line124: 'particular' ->'particularly'
- P5line128: 'such synergetic' ->'such a synergetic'
- P5line146: 'emssion' ->'emission'
- P7line171: 'each individual landfill' -> 'each landfill'
- P7line186: 'in Madrid area' -> 'in the Madrid area'
- P7line191: 'which brings error' -> 'which brings an error'
- P8line202: 'Due to its coarser spatial resolution the TROPOMI XCH4' -> add a ',' after resolution
- P8line212: 'This value fits well to' -> 'This value fits well with'

P9line228: 'as attenuated signal' -> 'as an attenuated signal'

P9line229: 'a time period of' -> 'a temporal window of'

P10line237:'during some years after sealing' -> 'for years after sealing'

P10line243: 'To better representing' -> 'To better represent'

Pllline276: 'fusiform-shape plumes' - >' fusiform-shaped plumes'

P12line283:' is due to noise of' -> 'is due to the noise of '

P12line295:' CH4 has a relatively longer lifetime than NO2' -> 'CH4 has a longer lifetime as compared to NO2'

P16line333: 'yields emission rates of close to' -> 'yields emission rates close to'

P18line374: 'derived from source location' -> 'derived from the source location'

P19line396: 'are indeed representative for' -> 'are indeed representative of '

P19line397: 'CH4 has a long lifetime' -> 'CH4 is a long-lived gas'

P20line423: 'As outlook, this methodology...' -> 'This methodology...'

P25 A-3: "personal communication of Omaira García" Why do you write this? If I understand correctly, Omaira García is one of the co-author.

P26 Eq8. I would like to see a table or an expansion for the x BG vector

P26 Eq 8. How do you calculate the K\_BG? Perturbation? If yes, then how do you choose the perturbation size? How does the size affect the result?

P26line510: The authors mentioned that  $K_BG^*$  is the same as  $K_BG$ , but set to zero for observations where the wind data suggest a significant impact of the  $CH_4$  plume on the satellite data. What do you mean by "a significant impact"?

P26line513: I guess the authors is talking about x\_BG instead of K\_BG?

**P26. Eq9: It is also confusing me that you wrote y\_BG in Eq.8 but y in Eq.9 without any explanation in the text.**

P26 line 519: How do you create the Sy,n and Sa matrices? If I understand correctly, these are key parameters for your y\_BG calculation.

Although the authors said that" The matrix Sy, n stands for the noise covariance of the satellite data ", where are the noise of the satellite measurements come from? from the satellite L2 data?

For Sa "with a very low constraint value for the coefficient determining the constant and higher constraint values for the other coefficients". What do you mean 'a very low'? What are the diagonal values for each retrieval parameter?

---

## Author Response (AR2)

Dear Prof. Dr. Landulfo,

thank you very much for handling our manuscript.

Based on the comments from REF#1, we have thoroughly modified the abstract and introduction, shortened the manuscript and made it more specific to the objectives of this study. We have moved two subsections ($CH_4$ background signal and fitting of $CH_4$ emission rates) to the appendix, and deleted some technical parts in the main text.

Response to Referee #1

We would like to thank the reviewer #1 for taking the time to review this manuscript again and provide valuable and constructive feedback that further improved the manuscript. We are very sorry that our prior modifications did not fulfill the reviewer's requirements. Below, we address the list of points raised by the referee. All the points one-by-one raised by the reviewer are copied here and shown in bold text, along with the corresponding reply from the authors in plain text.

**1. It is not clear how this case study of Madrid landfills is of sufficient general scientific interest to justify publication in ACP.**

Our study develops a novel method to estimate the $CH_4$ emission rates of landfills in metropolitan areas by using satellite and ground-based observations.

In case of the study region (Madrid) we used for demonstrating our method, all emission rates estimated from the different observations are significantly larger than the emission rates provided via the official Spanish Register of Emissions and Pollutant Sources. We expect that inventories in other parts of the world are also underestimating these kinds of emissions.

From the global perspective, the $CH_4$ emissions from landfills are significant contributions to anthropogenic emissions. We added several references in the introduction, showing the importance of estimating emissions from landfills.

**2. The paper is far too long. For example, the introduction starts with platitudes about the global methane budget that have little to do with what the paper is about. There is a lot of anecdotal detail about the results that may belong in a technical report to the city of Madrid but not in a scientific paper.**

We have thoroughly modified the abstract and introduction, shortened the manuscript and made it more specified on the objectives of this study. We have moved two subsections ($CH_4$ background signal and fitting of $CH_4$ emission rates) to the appendix, and deleted some technical parts in the main text.

**3. The English suffers from wordiness, bad grammar, and poor style.**

We have checked the grammar again and further revised the language of the manuscript. The final version will undergo language proofreading by the Copernicus editing group before publication.

**4. I am not convinced of the quality of the TROPOMI data, and the authors have done nothing to allay the concerns of my original review.**

We do not understand the concern of the referee with respect to the quality of the TROPOMI data. The TROPOMI data have been successfully validated and applied in our study for estimating emissions from landfills. We even demonstrated a reasonable agreement between the emission rate obtained from TROPOMI and the rate obtained from independent COCCON observations.

The TROPOMI $XCH_4$ has been validated with TCCON (-3.4 ± 5.6 ppb) and GOSAT (-10.3 ± 16.8 ppb) by Lorente et al., (2021), whose results is added to Sec.2.1.2. The mean bias between TROPOMI and COCCON is 2.7 ± 13.2 ppb, which is below the absolute bias between TROPOMI and TCCON. This information is also added to Sec.3.1. The following statements have been added to the manuscript:

Sec.2.1.2.: "This study uses the TROPOMI data set of $XCH_4$ from Lorente et al., (2021), for which an updated retrieval algorithm was implemented to obtain a data set with less scatter. This updated $XCH_4$ has been demonstrated to be in good agreement with TCCON (-3.4 ± 5.6 ppb) and GOSAT (-10.3 ± 16.8 ppb), with a bias and precision below 1%."

Sec.3.1.: "The mean bias in $XCH_4$ between TROPOMI and COCCON is 2.7 ± 13.2 ppb, which is below the absolute bias between TROPOMI and TCCON (3.4 ± 5.6 ppb, Lorente et al., 2021). The higher scatter of the validation with COCCON reflects the shorter temporal and spatial collocation, but the agreement It indicates that TROPOMI data have good quality and a low bias."

---

## Author Response (AR4)

Response to Referee #1

We would like to thank the reviewer #1 for taking the time to review this manuscript and provide valuable and constructive feedback that have improved the manuscript.

In this author comment all the points one-by-one raised by the reviewer are copied here and shown in bold text, along with the corresponding reply from the authors in plain text. We will go through our manuscript text and figures and try to shorten the paper where appropriate and we will submit a revised and restructured version of the manuscript when the second referee comment becomes available (for including all suggestions).

1. **The Introduction takes a very long time to get to the point, and at the end we still don't have a clear statement of the problem. Is the paper to quantify emissions from Madrid, or from three landfills, or is this the same thing? Why should we care about Madrid? What is actually new here?**

Sorry for this vaguely-referred information. We will make this clearer in the introduction. We estimate emissions from waste disposal sites as stated in the title of the paper. These three landfills are so close to the Madrid metropolitan area that they can be considered as city sources, and they are the most significant (also the unique significant) $CH_4$ sources in Madrid area. Meanwhile, these together disposal sites are rather strong source as compared with the inventory in Madrid. The metropolitan cities are continuously growing due to population movements, industries, etc., and thus, more and more cities incorporate landfills (and other $CH_4$ potential sources) into their limits/influential areas. This study might be also interesting for different big/medium cities and the method can be applied there.

2. **Section 2.1.2: give reference for the specific TROPOMI product that you are using. TROPOMI has a very low success rate (3% globally), is that an issue here? Or is Madrid sunny enough?**

(1) The specific TROPOMI products are from Lorente et al. (2021)*. We will add this to Section 2.1.2. according to the referee's comment.

(2) Yes, you are right that TROPOMI globally has a low success rate. This is more problematic at high latitudes and in winter. Southern Europe, like Madrid is pretty sunny and the weather during the field campaign was sunny as well.

The TROPOMI data with high quality (qa=1.0) used in this study are further collocated with the IASI data, covering the time period from November 2017 to October 2020. There are nearly 29,000 measurements. Although these observations are from many different days they can be used consistently for the local emission estimates, because we remove the daily varying background signal. This background removal is very important and allows for using data from many different days for achieving good coverage. The multi-year data set provides a large and consistent observational dataset (horizontal: for the whole area around Madrid; temporal: for different wind regimes) for studying the local emissions.

*Lorente, A., Borsdorff, T., Butz, A., Hasekamp, O., aan de Brugh, J., Schneider, A., Wu, L., Hase, F., Kivi, R., Wunch, D., 740 Pollard, D. F., Shiomi, K., Deutscher, N. M., Velazco, V. A., Roehl, C. M., Wennberg, P. O., Warneke, T., and Landgraf, J.: Methane retrieved from TROPOMI: improvement of the data product and validation of the first 2 years of measurements, Atmos. Meas. Tech., 14, 665–684, https://doi.org/10.5194/amt-14-665-2021, 2021.

3. **Section 2.1.3: this combined IASI+TROPOMI TXCH$_4$ product is probably totally dominated by TROPOMI information in the PBL, which is what matters here. So how is it independent from TROPOMI?**

TROPOMI measures column integrated methane (XCH$_4$). XCH$_4$ is affected by tropospheric CH$_4$ concentrations but also strongly by the altitude of the tropopause (a high tropopause causes low stratospheric contributions to XCH$_4$ and thus high XCH$_4$ values, a low tropopause vice versa). One possibility to avoid this contribution that affects the study of lower tropospheric CH$_4$ emissions is to remove the background and work only with anomalies. If one assumes that the background captures all the tropospheric background and in addition the stratospheric contribution, the anomaly contains the interesting signal. However, if the background calculation misses some of the stratospheric contribution signals, the interpretation using the anomaly for investigating local emission can lead to large errors. For instance, it might be that winds from the north-east and south-west are somehow correlated with tropopause altitudes (because both wind directions and tropopause altitudes have a seasonal cycle) and if this seasonal cycle is not well resolved in the background, the wind-assigned anomaly can have an artificial signal that comes from the stratosphere.

IASI has good information about the stratospheric contribution and combining IASI with TROPOMI allows for the generation of a product (TXCH$_4$) that is largely unaffected by the stratospheric contribution. Consequently, the risk for the TXCH$_4$ anomalies to be affected by the stratospheric contributions is much lower than for XCH$_4$. We calculate the emission for XCH$_4$ and for TXCH$_4$ and get similar emission rates for both data sets. This means our background correction is working correctly already for XCH$_4$. Furthermore, we repeat the calculations for the upper tropospheric and stratospheric CH$_4$ (UTSXCH$_4$, information from the IASI data) and found that there is no emission signal. All these prove that the emission signals we observe are not by accident, instead they come from local surface emissions and that our method for background calculation together with the wind-assigned anomaly method is able to detect these signals correctly in the XCH$_4$ as well as in the TXCH$_4$ data.

**4. Figure 1 is very difficult to read.**

Thanks for the comment. The figure is redone as below:

[Figure]

**5. Section 2.3: this section is very confusing because it is not clear what the authors are trying to optimize. Are the 'daily plumes' for the individual landfills? Are they summed over the three landfills? Are the three landfills treated as a single plume? The cone model is surely wrong for instantaneous plumes but is reasonable for time-averaged plumes, which is what is fitted but it takes the paper a while to explain this.**

Each individual landfill is considered as an individual point source. The daily plumes from the individual landfills are super-positioned to have a total daily plume. The final results we provide are emission rates averaged for the whole three years.

**6. Line 206: I don't get the point about seeking an analogy with $NO_2$. The landfills don't emit $NO_x$, $NO_x$ is an area source, and the decay of the $NO_2$ plume is by oxidation rather than dilution into background.**

Yes, you are right that no $NO_x$ is emitted from the landfills (while the Madrid metropolitan area is a strong source of $NO_2$). The usage of $NO_2$ in this study is to check if our method is reliable. The $NO_2$ is exploited as a tracer for atmospheric transport, offering sufficient chemical lifetime for forming a nice plume structure, which is a simple target gas for TROPOMI. $NO_2$ is a suitable (approximately stable) tracer for qualitatively demonstrating the method developed for the wind-assigned anomaly. We do not correctly consider the photochemical loss of $NO_2$, and the demonstration is not intended to provide a high-quality quantitative analysis of the $NO_2$ source strength (a refined model including $NO_2$ decay would generate slightly reduced outer plume lobes. The lifetime of $NO_x$ should be in the order between 6.3h during night and 29h during daytime in winter (Kenagy et al, 2018) and 5.9h in summer (Shah et al., 2020)). Our intention is simply to check the implementation of our approach and the data we feed into the simulation: if our wind data or plume dispersion modelling would be incorrect, we would not be able to reasonably reproduce the plume properly in our model runs.

* Kenagy, H. S., Sparks, T. L., Ebben, C. J., Wooldrige, P. J., Lopez-Hilfiker, F. D., Lee, B. H., Thornton, J. A., McDuffie, E. E., Fibiger, D. L., Brown, S. S., Montzka, D. D., Weinheimer, A. J., Schroder, J. C., Campuzano-

Jost, P., Day, D. A., Jimenez, J. L., Dibb, J. E., Campos, T., Shah, V., Jaeglé, L. and Cohen, R. C.: NOx Lifetime and NOy Partitioning During WINTER, J. Geophys. Res. Atmos., 123(17), 9813–9827, doi:https://doi.org/10.1029/2018JD028736, 2018.

Shah, V., Jacob, D. J., Li, K., Silvern, R. F., Zhai, S., Liu, M., Lin, J., and Zhang, Q.: Effect of changing NOx lifetime on the seasonality and long-term trends of satellite-observed tropospheric NO2 columns over China, Atmos. Chem. Phys., 20, 1483–1495, https://doi.org/10.5194/acp-20-1483-2020, 2020.

**7. Wind speed is denoted v in the text, w in Figure 2.**
Many thanks to point out this mistake. We corrected it.

[Figure]

**8. Equation (9): not clear how you get y_BG**

For clarification, Eq. 9 $y_{BG}$ should be replaced by $y$ (the original satellite data $y=y_{BG}+y_{plume}$, see Eq. 7). We use $y$ to estimate the coefficients that describe the background. Information from $y$ is only used when the observations are not affected by the plume. In that case $\mathbf{K_{BG}}^*$ and thus $\mathbf{G_{BG}}$ are zero. On the other hand, $\mathbf{K_{BG}}^*$ (and thus $\mathbf{G_{BG}}$) is set to zero whenever $y_{plume}=0$. Then $y_{BG}=y$ and Eq. 9 is correct as it. But in order to make it clearer we will write in Eq. 9 $y$ instead of $y_{BG}$.

**9. Section 2.3: there are many uncertainties in the procedure for inferring emissions. How can it be validated? An obvious way would be to use the independent COCCON observations to evaluate the posterior concentrations resulting from the TROPOMI inversion.**

Estimated emission rates can include large uncertainties. In this study, the uncertainty of the estimated emission rates derived from TROPOMI XCH$_4$ ($7.4\times10^{25} \pm 6.4\times10^{24}$ molec s$^{-1}$) based on our method is about 9% ($6.4\times10^{24}/7.4\times10^{25}$). Please note that this is only the uncertainty due to the background uncertainty. Figure 10 shows sensitivity analysis due to wind, emission source, and opening angle, and reveals that there are other important uncertainty sources.

The referee gives a very good suggestion that COCCON is an independent data to evaluate the results. We applied this strategy and tried to estimate the emission rate from COCCON measurements on October 4 (Section 3.1), when a significant enhancement was observed by the downwind-side COCCON SN69. The estimated rate is about $3.7\times10^{25}$ molec s$^{-1}$. This value is about half of that derived from TROPOMI XCH$_4$ or combined TXCH$_4$, which is a plausible match, because the satellite covers the complete area, whereas the COCCON plume observation primarily detects the emission from a single nearby landfill.

**10. Figure 3: the agreement between TROPOMI and COCCON in that Figure strikes me as very poor, despite the authors' claim to the contrary. I'm not surprised by this in view of the known TROPOMI biases, but it undermines confidence in the results of the TROPOMI inversion. The paper goes on about the problems on Sept 25 and Oct 4 but that seems anecdotal and those two days don't seem any worse than the rest of the population in Figure 3.**

We do not recognize an apparent bias between COCCON and TROPOMI (in Fig 3 we see some scatter, but the whole ensemble follows the 1:1 line quite well). Larger variations of $XCH_4$ in a metropolitan area containing localized sources as Madrid are to be expected (which will induce some scatter between the datasets, because the spatial resolution of the space-based sensor is much lower). Studies under background conditions have revealed very good agreement and low bias between COCCON and TROPOMI (Tu et al., 2020*). Moreover, it needs to be emphasized that our conclusions on emission strengths depend on averaged values of observed *gradients* in each dataset, so a general bias due to a calibration mismatch between satellite and ground-based would remain largely without effect.

Tu, Q., Hase, F., Blumenstock, T., Kivi, R., Heikkinen, P., Sha, M. K., Raffalski, U., Landgraf, J., Lorente, A., Borsdorff, T., Chen, H., Dietrich, F., and Chen, J.: Intercomparison of atmospheric CO2 and CH4 abundances on regional scales in boreal areas using Copernicus Atmosphere Monitoring Service (CAMS) analysis, COllaborative Carbon Column Observing Network (COCCON) spectrometers, and Sentinel-5 Precursor satellite observations, Atmos. Meas. Tech., 13, 4751–4771, https://doi.org/10.5194/amt-13-4751-2020, 2020.

**11. Lines 329-330: how do we know that the 'COCCON instruments show a very good ability to detect the source'? No specific results or data from COCCON are shown.**

The COCCON SN69 was located in the northwest of the landfill Valdemingómez with a quite close distance (4.5 km). When the wind came from southeast, the COCCON SN69 was located downwind of the landfill and detected a significant plume (nearly up to 100 ppb on October 4, 2018, Figure 4), whereas the other COCCON sites did not observe any enhancements. Another example is October 1, 2018 (Figure A-2): the wind direction was north to northeast and the COCCON stations were not on the downwind side of the landfills, which resulted in no enhancement at any of the COCCON stations. The observations on different days were largely depended on the wind situation. The obvious downwind enhancement observed by the COCCON instruments demonstrates that they have the ability to detect the emissions of the source, which has been demonstrated in Kille et al. (2019*) as well.

* Kille, N., Chiu, R., Frey, M., Hase, F., Sha, M. K., Blumenstock, T., Hannigan, J. W., Orphal, J., Bon, D. and Volkamer, R.: Separation of Methane Emissions From Agricultural and Natural Gas Sources in the Colorado Front Range, Geophys. Res. Lett., 46(7), 3990–3998, doi:https://doi.org/10.1029/2019GL082132, 2019.

**12. Figure 8 is cryptic. What domain is shown? What are we learning from it?**

We have tried to visualize as good as possible the different steps of the data treatment. Figure 7 shows the time series of the different satellite data ($y$ from Eq. 7), their estimated background ($y_{BG}$ from Eq. 12), and the anomaly signal due to the local emissions ($y_{plume}$ from Eq. 13). Then Figure 8 shows the anomalies (i.e. $y_{plume}$) horizontally averaged for different wind directions. This data is used for the $\Delta$-calculations. Eq.15 captures both: the horizontal averaging according to the wind directions as well as the $\Delta$ calculations. The results of these calculations are then shown in Fig. 9. These $\Delta CH_4$ data are finally used to estimate the emission rates according to Eq. 18.

Furthermore, Figure 8 is very useful here, because it demonstrates that the $CH_4$ hotspots are south-east of Madrid and not in the center of the city. South-east of Madrid is where the waste disposal sites are located.

**13. Table 5: I don't see the relevance of this Table to the paper.**

This Table shows some results from other studies as a reference for our results. It helps to demonstrate that our results are reliable and lie in the reasonable range. The inventory only lists the active landfill cells and does not include the closed ones, which probably still emit for many years (Sánchez et al., 2019*). This is an additional argument for the relevance of the kind of work we are presenting here.

*Sánchez, C., de la Fuente, M. del M., Narros, A., del Peso, I. and Rodríguez, E.: Comparison of modeling with empirical calculation of diffuse and fugitive methane emissions in a Spanish landfill, J. Air Waste Manage. Assoc., 69(3), 362–372, doi:10.1080/10962247.2018.1541029, 2019.

Response to Referee #2

We would like to thank the reviewer #2 for taking the time to review this manuscript and provide valuable and constructive feedback that have improved the manuscript.

In this author comment all the points raised by the reviewer are copied here one-by-one and shown in bold text, along with the corresponding reply from the authors in plain text.

1. **The Introduction gives extensively credit on former works and is therefore also an important part, which should not be missing even it might not help the reader in understanding of the new technics.**

   **Therefore, I would ask for a kind of road map at the beginning of second section "method", before 2.1 which gives an idea, why and how the measurements and models are combined, or maybe it could be included directly at the end of introduction where the different sections are already mentioned.**

Here, we add two more sentences to describe our methodology in the last paragraph of the introduction:

"Section 2 describes our methodology. We calculate the difference of the satellite data maps for two opposite wind regimes (we refer to the resulting signals as wind-assigned anomalies). A simple plume model is applied to predict the wind-assigned anomalies for a chosen position and strength of a source. The results of our study are presented and discussed in Section 3……"

2. **TROPOMI+IASI**
   **2.1 You report two estimations in the abstract, which are slightly contra dictionary. The error in the estimation is higher than the error estimation from the combined retrieval, I imagine that this is more or less the effect that partial columns are constructed with more errors (Error of TROPOMI + error of IASI), but there should be another systematic error, which is reduced if the l tropospheric partial column anomaly is sub or over estimated. It would be easier for us readers, if the authors conclude on a single best estimation, and report the other option using just TROPOMI just in the results and discussion of the article.**

As stated correctly by the referee, the stratospheric contribution causes larger uncertainties when using the $XCH_4$ data. However, it is very difficult to estimate this error. In the paper, we do not estimate the uncertainty of the stratosphere contribution. Instead, we use combined data (IASI+TROPOMI, where the stratospheric contribution can be neglected) to show that our background removal method works well and removes the stratospheric effects, also in the $XCH_4$ data.

**2.2 The combination of IASI and TROPOMI assumes measurements of the same air mass, and therefore at the same time, but IASI comes around 10:00 and TROPOMI around 13:00, does this matter here, o IASI characterize more the background CH$_4$ and is therefore less critical. Maybe it would be interested in a discussion how the time difference between IASI and TROPOMI of some hours, might affect the combined retrieval.**

**Please also describe a bit more how much the estimation improves due the combination with IASI, I could imagine, that this advantage will increase if you use long term averages, so that the random error loses its importance in relation to improved sensitivity in the lower atmosphere.**

In Schneider et al. (2021b) we found that a temporal mismatch of up to four hours is no problem. The noise error is larger than the error introduced by combining two measurements that have a temporal mismatch of four hours. The mismatch error just increases slightly the statistical error budget.

Yes, the noise error is larger in the combined product (see Schneider et al., 2021b). And this error will get smaller, when more data are used for averaging. So, if the emission rate keeps constant over a long time, the estimation of these emission rate will become the better, the longer the time series is (the more data are used for averaging). However, in practice the emission rates might not be fully constant over time.

**3. Plume model**

**3.1 The authors realize their proper idea and do not have to cite other works, which might be partly similar ideas, but it would help some reader to understand it easier, which is similar to other approaches (up, downwind Rotating method, Gaussian's Law).**

The comment is very helpful to make reader better understand of our simple model. We try to explain our model more:

Our plume model is a simplified version of the Gaussian plume model (https://link.springer.com/chapter/10.1007/978-1-4757-4465-1_7, Figure 7-1). Our simple model treats a two-dimensional concentration field (without height being involved) generated by a point source (or an array of point sources) and the gas molecules in the downwind side are evenly distributed into a sector (not a Gaussian distribution) centered along the wind direction. The opening angle of the sector is the only free model parameter, which is adjusted in order to reasonably reproduce the observe NO$_2$ plume.

**3.2 Do I understand correctly the point, that the plume model is retrieved using the City centre of Madrid as central source, but applied to the source located at a different location (landfills). So orography should be similar or not be important. I would have been interested in the required conditions on the regional topography for this approach to be transfer also to other cities or waste deposals sites. And I also would like to know the typical mixing layer height in the area around Madrid.**

For $NO_2$ we use the Madrid city center as a central source for the plume model. However, we use three appropriately located point sources in case of $CH_4$. The a priori information about the $CH_4$ emissions from the landfills (Table 3) is taken from the Spanish Register of Emissions and Pollutant Sources. Each individual landfill is considered as a point source. The contributions from the individual landfills are super-positioned to generate a total daily plume.

The regional topography from Google map and the altitude derived from TROPOMI are presented as below. The mountain ranges locate along the NE-SW direction, which forces the wind flowing along this direction. We, however, do expect similar wind conditions in the city center and the landfills located nearby, as the topography is rather flat in this sub-region.

[Figure]

Our Spanish colleagues kindly compiled information on the Planetary Boundary Layer (PBL) height: the height of the PBL in Madrid, and its seasonal evolution, is the typical from continental areas. Figure below shows the averaged seasonal cycle of the height of PBL using meteorological radiosondes launched twice a day from the Madrid-Barajas station (WMO #08221) in the period 1981-2015. It is calculated using the Heffter method (Heffter JL. 1980. "Transport Layer Depth Calculations." Second Joint Conference on Applications of Air Pollution Meteorology, New Orleans, Louisiana). Note that the radiosonde launch station is the same used in the MEGEI-MAD campaign.

So, the noon PBL reaches typically altitudes of about 1300 m in wintertime and up to 2500 m in summertime. We discuss possible corrections due to increasing wind speed with altitude on our emission estimates further below.

[Figure]

**3.3** **The test using NO₂ from TROPOMI results in a lower emission rate compared to estimation from the literature. The authors identify the lifetime as the reason and maybe also the value in the literature might have errors, or might not be comparable, different time ....However a validation of the wind speed estimation from a 10 m altitude would be nice.**

**For CH₄ and its long lifetime the anomaly in the column should be definitively given by 1/d, as described by the equation 3.**
**The forward model contains the velocity and actually the quotient emission/velocity determine the concentration, or vice versa in the inversion. That is why the wind speed estimation might be so important. The sensitivity study using the wind at 10 m is very helpful and important, never the less I did not get completely get the point how you decided that 10 m is the best. I think the wind velocity you look for is the velocity you can multiply with the column anomaly to get the total CH₄ flux. Ad hoc I would take the average windspeed in the atmospheric boundary layer.**

**The COCCON sites at "Jose Echegaray" and "Barajas" have a distance of more or less around 8km, on 25 September 2018 (Figure 4) we see that anomalies en CH4 arrives around 1-2 hours later in Barajas than it appears in "Jose Echegaray".**

**So the velocity of propagation of this plume might be 8 km/2 h to 8 km/h or 1.11m/s to 2.2 m/s respectively, this fits actually very well to the assumed and modelled wind velocity in the plot below. Surely that is the intention showing the plot, but maybe it should be also explicitly be mentioned, that you can also use the 5 FTIR sites to validate the wind estimation strategy you have chosen, as here the interest in the effective flux of CH4 and it might be the best validate the velocity using the FTIR EM27 measurements of columnar CH4, so that you are independent of the vertical distribution of CH4.**

**EM27 measurements take typically a little bit less than a 1 minute, maybe you could do a cross correlation to retrieve the delay after interpolate data to a common 1 minute grid, and then include a point in the figure showing the wind speed or projection, but it is sufficient just to mention it.**

We thank the referee for the careful consideration concerning the wind. We fully agree that the limited quality of the available wind data is a significant source of uncertainty.

According to WMO (WMO, 2018), the measurement representative of the surface winds is the wind records at 10 m a.g.l. to avoid the roughness of surrounding terrain. In this sense, the winds at 10 m are usually taken as a proxy for surface emission estimations (e.g. Viatte et al., 2017), such as in the Madrid case. In addition, we chose ERA5 wind at 10 m, because it can be directly compared to the in situ wind observations at 10m at the Cuatro Vientos Airport, which helps to estimate the uncertainty introduced by the wind data.

The wind at ~500 m a.g.l (900 hPa) would be a more appropriate choice for transport modelling if we allow for vertical mixing of the plume within the PBL. As shown in the Table 1 and **Figure 1** below, there is a significant increase of wind speed with altitude. The ERA5 wind data at 10 m and ~500 m do not differ significantly concerning the wind direction, but the wind velocity increases with altitude. The wind speed increases by 60%, i.e. using ERA5 wind information at ~500 m instead of that at 10 m would yield 60% increase in the emission rate.

**Table 1:** ERA5 wind at 10 m and 100 m in TROPOMI overpass days.

| Wind direction range | 10 m | | ~500 m/900 hPa | |
|---|---|---|---|---|
| | Number of days in total (%) | Averaged wind speed ± standard deviation (m s$^{-1}$) | Number of days in total (%) | Averaged wind speed ± standard deviation (m s$^{-1}$) |
| NE / >315° or <135° | 28.4 | 2.3 ± 1.2 | 30.9 | 3.8 ± 2.1 |
| SW / 135° – 315° | 61.8 | 2.3 ± 1.4 | 56.7 | 3.6 ± 2.0 |

[Figure]

**Figure 1:** Wind roses for daytime (08:00 UTC – 19:00 UTC) for the ERA5 model wind at 10 m (left) and at ~500 m (900 hPa) (right).

It is really a good hint to mention that we should use the array of COCCON stations itself to validate the assumptions on the wind field. We will add this statement to our manuscript:

"These five COCCON stations can serve as an independent source of information for constraining the wind speed. For example, the distance between the Jose Echegaray and Barajas is about 10 km. The highest anomalies of XCH$_4$ arrived around 1.5 hours later at Barajas station than it appeared at the Jose Echegaray station on 25 September 2018, which indicates an averaged wind speed of 1.8 m/s. This value fits well to the ERA5 model wind velocity."

*WMO, Guide to Instruments and Methods of Observation Volume I –Measurement of Meteorological Variables, Report No. 8, ISBN 978-92-63-10008-5, Geneva, Switzerland, 2018.*

*Viatte, C., Lauvaux, T., Hedelius, J. K., Parker, H., Chen, J., Jones, T., Franklin, J. E., Deng, A. J., Gaudet, B., Verhulst, K., Duren, R., Wunch, D., Roehl, C., Dubey, M. K., Wofsy, S., and Wennberg, P. O.: Methane emissions from dairies in the Los Angeles Basin, Atmos. Chem. Phys., 17, 7509–7528, https://doi.org/10.5194/acp-17-7509-2017, 2017.*

**4.  Page 2, Line 37:**

**4.1 "The wind-assigned plume method is also applied to the tropospheric and upper tropospheric/stratospheric column averaged CH4 mixing ratio products (in the following referred to as TXCH4 and UTSXCH4) derived from a-posteriori merged Infrared Atmospheric Sounding Interferometer (IASI) profile and TROPOMI total column data." Maybe you could split the sentence into 2 and find somehow a different description, which is easy to understand:**
**For the CH4 emission estimation, the wind-assigned plume method is applied to the lower tropospheric methane /dry air column ratio (TXCH) of the combined TROPOMI Infrared Atmospheric Sounding Interferometer (IASI) Product. TXCH4 and the upper tropospheric/stratospheric column averaged CH4 mixing ratio (UTSXCH4) are derived from a-posteriori merged Infrared Atmospheric Sounding Interferometer (IASI) profile and the TROPOMI total column data.**

We modify the sentence according to referee's comments:

"For the CH$_4$ emission estimation, the wind-assigned plume method is applied to the lower tropospheric CH$_4$/dry air column ratio (TXCH$_4$) of the combined TROPOMI + IASI (Infrared Atmospheric Sounding Interferometer) product. TXCH$_4$ and the upper tropospheric/stratospheric column averaged CH$_4$ mixing ratio (UTSXCH$_4$) are derived from a-posteriori merged IASI profile and the TROPOMI total column data."

**4.2 Does it make sense to apply the method to the UTSXCH4 Product? As mentioned earlier I would just use TXCH4 in the abstract and conclusion.**

The very low emission rate derived from the UTSXCH$_4$ demonstrates that all is working well: from TXCH$_4$ we get the same as for XCH$_4$ (if background is carefully removed). And in UTSXCH$_4$ we see no emission signal (i.e. the background removal and the calculation of wind-assigned anomalies introduce no artificial signals).

**5. Line 40: The first sentence is a bit redundant, maybe you could drop here the sentence "Based on the NE and SW wind fields, we developed a simple plume model locating the source at three waste disposal sites east of Madrid for CH4." or move it to line 37.**

We remove this sentence according to the referee's comment.

**6. Line 44: "day and. All em…**
**Maybe you could rewrite the sentence which is also not clear." The COCCON observations indicate a weaker CH4 emission strength of around 3.7×1025 molec s-1 from local source (near to the Valdemingómez waste plant) in accordance with observations in a single day and. "Please write very clear if COCCON is used to estimate an independent "local source", if this source is part and included in the Tropomi estimation given quantitatively in the line above. Please state if TROPOMI and COCCON based estimations are contradiction, complementary and/or consistent. If you want, you might move the last sentence to line 25, as it is finally the most important finding.**

The comment helps us to explain the results clearer. We change this sentence to:

"COCCON observations are investigated to estimate the local source as an independent method. The COCCON observations indicate a weaker $CH_4$ emission strength of around $3.7\times10^{25}$ molec s-1 from local source (near to the Valdemingómez waste plant) in accordance with observations in a single day. That this figure is lower than the one derived from the satellite observations is a plausible result, because the analysis of the satellite data refers to a larger area, covering further emission sources in the study region, whereas the signal observed by COCCON is generated by a nearby local source."

**7. Figure 8,9:**
**Please use a fixed color scale in Fig 9 and maybe also in Fig 8. The modelled XCH4 for total column, lower and upper troposphere seems to be exactly the same in fig 9 b,e,h. ….. Maybe when you talk about emission rates line 474 and line 477 you cloud use CH4 and not the dry air mol fraction.**
**XCH4-> CH4**

To remove unwanted signals due to validations of ground pressure and atmospheric humidity we estimated the emission rates from total and partial column averaged dry air mixing ratios and not directly from total column amounts. The color scales in Figure 8 and 9 are chosen according to the typical dry air columns. The dry air total column is typically by a factor of 1.9 larger than the dry air tropospheric partial column. And it is typically by a factor of 2.1 larger than the dry air upper tropospheric/stratospheric partial column. These factors explain the different color scales.

Response to Referee #3

We would like to thank the reviewer #3 for taking the time to review this manuscript and for providing valuable and constructive feedback that helped us to further improve the manuscript.

In this author comment all the points raised by the reviewer are copied here one-by-one and shown in bold text, along with the corresponding reply from the authors in plain text.

**Major comments**

**1. Reconsider the structure of the introduction. For the moment, I cannot capture what is the current status of the CH4 emission research from the ground-based and space-based measurements, what are the key issues, and what the authors will do in this work to solve/improve the issues. For example, I would like to suggest moving some texts in Section 2.1 and Section 2.2 to the introduction. Only keep the data description/technical part in Section 2.**

Thank you for the comments. We followed your suggestions by adding more information to introduce the ground-based and space-based measurements. Therefore, we moved the respective text parts from Section 2.1 and 2.2 to the introduction and added additional references.

**2. In the abstract: "As CH4 emission strength we estimate $7.4 \times 10^{25} \pm 6.4 \times 10^{24}$ molec s-1 from the TROPOMI XCH4 data and $7.1 \times 10^{25} \pm 1.0 \times 10^{25}$ molec s-1 from the TROPOMI+IASI merged TXCH4 data." Why the uncertainty derived from the TROPOMI+IASI TXCH4 data is larger than that from the TROPOMI data? I thought that the advantage of using TROPOMI+IASI is to obtain more information in the troposphere so that users can reduce the uncertainty.**

It is not possible to have the $TXCH_4$ by either the TROPOMI or IASI product individually. The synergetic combination of TROPOMI and IASI enables us to detect tropospheric $CH_4$ independently from the upper tropospheric/stratospheric $CH_4$. Because the TROPOMI+IASI $TXCH_4$ product is not influenced by the varying tropopause, it is more sensitive to the tropospheric variations than the $XCH_4$ data. On the other hand, the merged product has a larger noise error: (1) because the vertical distribution of $CH_4$ is in general much more difficult to measure than the total column of $CH_4$ and (2) because we derive the vertical distribution by considering two independent measurements, each with its own noise error. With the current data availability, we estimate that the emission rates obtained from the $TXCH_4$ data have a slightly higher uncertainty than the emission rates we obtain from the $XCH_4$ data. This might change for a larger number of data points (e.g. by using data from more years or by applying the method to IASI and TROPOMI successors on the upcoming METOP-SG satellite, which offers much more collocated observations).

However, we would like to point out that in our study using $TXCH_4$ data in addition to $XCH_4$ data nicely documents the robustness of the method. Important for a correct estimation of the

emission is the correct removal of the methane background signal. For $XCH_4$ the stratospheric and the tropospheric background have to be removed. For $TXCH_4$ only the tropospheric background has to be removed. In our study we use $XCH_4$ and $TXCH_4$ data sets. Figure 7 shows the rather different background signal of $XCH_4$ and $TXCH_4$. Despite this difference we estimate very similar emission rates from both data sets (the emission rate uncertainties using $XCH_4$ or $TXCH_4$ are insignificant compared to the estimated emission rates). This proves that our method gives robust results even when using data with rather different background signals.

**Minor/technical comments:**

**P2line36. It is confused with "that this". Please reword this sentence" That this strength is lower than the one derived from the satellite observations is a plausible result."**

corrected.

**P2line45: "are to" -> "are"**

corrected.

**P2line49: 'while' -> 'in which'**

corrected.

**P2line49: What does '~55% uncertainty' mean? Please clarify it in the text.**

clarified. It means that in the global uncertainty share the landfills own 55%.

**P2line53: 'space borne' -> 'space-borne'. As you use both ground-based and space-based measurements in the study, why do you only highlight the space-based data here?**

corrected. Thanks for pointing it out. The ground-based data should be also highlighted here.

**P2line62: 'TCCON' for the first time, please write down the full name**

corrected. The full name is "Total Carbon Column Observing Network" and it has been added to the text.

**P3line70: 'column-average' -> 'column-averaged'**

corrected.

**P3line81: 'The Bruker EM27/SUN' -> 'A Bruker EM27/SUN'**

Here we mention the specific instrument and we think "the" is a proper word.

**P4line115: 'we apply a strict quality control' - > 'we apply strict quality control'**

corrected.

**P4line124: 'particular' ->'particularly'**

corrected.

**P5line128: 'such synergetic' ->'such a synergetic'**

corrected.

**P5line146: 'emssion' ->'emission'**

corrected.

**P7line171: 'each individual landfill' -> 'each landfill'**

Here we would like to emphasize that each landfill is considered as an independent source. Therefore, we use "each individual" here.

**P7line186: 'in Madrid area' -> 'in the Madrid area'**

corrected.

**P7line191: 'which brings error' -> 'which brings an error'**

corrected.

**P8line202: 'Due to its coarser spatial resolution the TROPOMI XCH4' -> add a ',' after resolution**

corrected.

**P8line212: 'This value fits well to' -> 'This value fits well with'**

corrected.

**P9line228: 'as attenuated signal' -> 'as an attenuated signal'**

corrected.

**P9line229: 'a time period of' -> 'a temporal window of'**

corrected.

**P10line237:'during some years after sealing' -> 'for years after sealing'**

corrected.

**P10line243: 'To better representing' -> 'To better represent'**

corrected.

**Pllline276: 'fusiform-shape plumes' - >' fusiform-shaped plumes'**

corrected.

**P12line283:' is due to noise of' -> 'is due to the noise of'**

corrected.

**P12line295:' CH4 has a relatively longer lifetime than NO2' -> 'CH4 has a longer lifetime as compared to NO2'**

corrected.

**P16line333: 'yields emission rates of close to' -> 'yields emission rates close to'**

corrected.

**P18line374: 'derived from source location' -> 'derived from the source location'**

corrected.

**P19line396: 'are indeed representative for' -> 'are indeed representative of'**

corrected.

**P19line397: 'CH4 has a long lifetime' -> 'CH4 is a long-lived gas'**

corrected.

**P20line423: 'As outlook, this methodology…' -> 'This methodology…'**

corrected.

**P25 A-3: "personal communication of Omaira García" Why do you write this? If I understand correctly, Omaira García is one of the co-author.**

Thank you for pointing it out. We will remove this text.

**P26 Eq8. I would like to see a table or an expansion for the x_BG vector**

$x_{BG}$ is a vector, consisting of the coefficients for each component of the background model. The components of the background model are described in line 513-516 and more detailed information is below.

**P26 Eq 8. How do you calculate the K_BG? Perturbation? If yes, then how do you choose the perturbation size? How does the size affect the result?**

$K_{BG}$ is a matrix, where each row represents an individual satellite observation and each column a component of the background model. The first component of the background model is a constant, thus the entries in the first column of $K_{BG}$ are 1.0 for each row (for each observation). The second component of the background model represents the linear of $CH_4$, thus the second column of $K_{BG}$ is the time (t) when the respective observation has been made (each row might have a different observation time). The next six columns of $K_{BG}$ are for the seasonal cycle and the entries are ($\sin(2\pi t/365)$ and $\cos(2\pi t/365)$ for the $\frac{1}{year}$ frequency, $\sin(4\pi t/365)$ and $\cos(4\pi t/365)$ for the $\frac{2}{year}$ frequency, and $\sin(6\pi t/365)$ and $\cos(6\pi t/365)$ for the $\frac{3}{year}$ frequency). Then for fitting the daily anomaly there are further columns, where each column represents data from a single day. Rows (i.e. observations) that represent this day have entry 1.0, all others have entry 0.0. For fitting the horizontal anomaly (which is constant in time) there are columns, where each column represents a horizontal location (latitude × longitude resolution of 0.1° × 0.135°). Each row (i.e. observation) that represents this location has entry 1.0 and all other rows have entry 0.0.

**P26line510: The authors mentioned that K_BG* is the same as K_BG, but set to zero for observations where the wind data suggest a significant impact of the CH4 plume on the satellite data. What do you mean by "a significant impact"?**

We calculate the plume signal according to Sect. 2.3, i.e. for each observation we have a

theoretical plume signal. All observations for which we expect a plume signal being beyond the 75% percentile of all plume signal values is defined to be significantly affected by the plume. The 75% percentile is chosen empirically. It ensures that the background calculation is not significantly affected by the plume and that there are still sufficient observations available for estimating the background in a robust manner.

**P26line513: I guess the authors is talking about x_BG instead of K_BG?**

Here we refer to $\mathbf{K_{BG}}$. Maybe we clarify this by writing: "$\mathbf{K_{BG}}$ is a Jacobian matrix where each row represents an individual satellite observation and each column a component of the background model. The background model considers a smooth background, which is …". This has been added to the revised manuscript.

**P26. Eq9: It is also confusing me that you wrote y_BG in Eq.8 but y in Eq.9 without any explanation in the text.**

$\mathbf{K_{BG}}^*$ (and thus $\mathbf{G_{BG}}$) is set to zero whenever $y_{plume}=0$. This means for Eq. 9 we can replace $y_{BG}$ by $y$, i.e. use $y$ in Eq. 9 instead of $y_{BG}$, which one would expect from Eq. 8. Actually, $y_{BG}$ is what we want to estimate (see Eq. 12). We see the point of the referee and will add after Eq. 9 the following explanation: "Because $\mathbf{K_{BG}}^*$ (and thus $\mathbf{G_{BG}}$) is set to zero whenever $y_{plume}=0$, we can use in Eq. 9 $y$ instead of $y_{BG}$."

**P26 line 519: How do you create the Sy,n and Sa matrices? If I understand correctly, these are key parameters for your y_BG calculation.**

**Although the authors said that" The matrix $S_{y,n}$ stands for the noise covariance of the satellite data ", where are the noise of the satellite measurements come from? from the satellite L2 data?**

**For Sa "with a very low constraint value for the coefficient determining the constant and higher constraint values for the other coefficients". What do you mean 'a very low'? What are the diagonal values for each retrieval parameter?**

Thanks for these important comments – we should have been more explicit on these points.

Both $\mathbf{S}_{y,n}$ and $\mathbf{S}_a$ are diagonal matrixes. For $\mathbf{S}_{y,n}$ the noise comes from the satellite measurements, i.e. the $XCH_4$ precision of the satellite data.

The diagonal values of $\mathbf{S}_a$ are $(20\ ppb)^2$ for the coefficients representing the seasonal cycle, the daily anomalies and the horizontal anomalies. The diagonal value of $\mathbf{S}_a$ that represent the constant $CH_4$ values are set to $(10000\ ppb)^2$ and the diagonal value of $\mathbf{S}_a$ that represents the linear temporal increase is set to $(1\ ppb/day)^2$.

Concerning the uncertainty treatment (Eq. 11, 14, 20, 21) we improved the related text. In addition, please note that in Eq. 12, 13, 14 and 20 there were typos, where $\mathbf{K}$ should be $\mathbf{K_{BG}}$. This has been corrected in the revised manuscript.